# Separate brainstem circuits for fast steering and slow exploratory turns

Lulu Xu[1,2], Bing Zhu[1,2], Zhiqiang Zhu[1,2], Xingyu Tao[1,2], Tianrui Zhang[1,2], Abdeljabbar El Manira ●[3] ✉ & Jianren Song ●[1,2,3] ✉

Locomotion requires precise tuning of descending commands to scale turning movements, such as rapid steering during prey pursuit or shallow turns during exploration. We show that these two turn types are governed by distinct brainstem circuits. The rapid steering circuit involves excitatory V2a and inhibitory commissural V0d neurons, distributed across different brainstem nuclei. These neurons are coupled via gap junctions and activated simultaneously, ensuring rapid steering through asymmetrical activation of spinal motor neurons. The recruitment of this circuit correlates more with the degree of direction change than with locomotor frequency. Steering neurons are, in turn, controlled by a subset of V2a neurons in the pretectum, activated by salient visual input. In contrast, slow exploratory turns are governed by a separate set of V2a neurons confined to fewer brainstem nuclei. These findings reveal a modular organization of brainstem circuits that selectively control rapid steering and slow exploratory turning during locomotion.

Locomotion is a universal motor behavior within the animal kingdom, guiding movements towards desired destinations[1–4]. Switching between different locomotor programs, from exploration to predation and escape, is a common feature of vertebrate behavior[5–9]. This requires a high degree of flexibility for precise and rapid adjustments in speed and direction[1,5,10]. Such flexibility enables necessary adaptations crucial for pursuing salient stimuli, such as moving prey, which requires rapid steering turns and quick changes in direction. Conversely, routine shallow turns facilitate exploration, seamlessly allowing trajectory adjustments during slower locomotion[11,12].

Brainstem projecting neurons provide descending commands to initiate locomotion, which are then transformed into rhythmic locomotor patterns by spinal circuits[3,5,7,13–15]. These spinal circuits, owing to their modular organization, sequentially recruit motoneurons from slow to intermediate and then fast locomotion as the speed increases[16–22]. Emerging evidence suggests that locomotor speed is also encoded by the brainstem, with distinct circuits driving slow and fast locomotor movements[23]. Furthermore, turning gait asymmetry in mammals has been shown to be mediated by excitatory descending

neurons in the medulla expressing the transcription factor Chx10 (Vsx2, V2a-Gi neurons)[24]. The activity of these neurons is in turn regulated by the basal ganglia through a specific pathway via a pontine nucleus, oral part (PnO)[25]. Similarly, Chx10 expressing neurons have been shown to be involved in turning in zebrafish[26]. In addition to speed changes, locomotion inherently involves trajectory changes that require precise tuning of the descending commands to scale turning movements in a task-specific manner, producing either rapid steering turns or routine shallow turns associated with exploration in an open environment[5,12]. A central question that has proven challenging to address is whether shared or segregated brainstem circuits encode rapid steering turns versus slow exploratory turns. Addressing this question requires accessing brainstem circuits at a single-cell resolution in behaving animals in vivo and understanding how they integrate within spinal circuits to produce different turn patterns for the desired behavioral outcomes.

In this study, we leverage the experimental accessibility of larval zebrafish, coupled with detailed behavioral, electrophysiological, and imaging analyses, to uncover the brainstem circuits encoding rapid

[1]Shanghai Key Laboratory of Anesthesiology and Brain Functional Modulation, Clinical Research Center for Anesthesiology and Perioperative Medicine, Translational Research Institute of Brain and Brain-Like Intelligence, Shanghai Fourth People's Hospital, School of Medicine, Tongji University, Shanghai, China. [2]Center for Brain and Spinal Cord Research, Tongji University, Shanghai, China. [3]Department of Neuroscience, Karolinska Institute, Stockholm, Sweden. ✉e-mail: abdel.elmanira@ki.se; song.jianren@tongji.edu.cn

steering and slow exploratory turns. Our findings reveal that the two turning patterns are governed by distinct, non-overlapping brainstem circuits. The rapid steering circuit is limited to a few neurons that are widely distributed across multiple brainstem nuclei. In contrast, the circuit controlling slow exploratory turns resides in fewer nuclei. The steering turn circuit consists of four pairs of V2a neurons distributed across three different nuclei, along with a pair of commissural V0d neurons. Together these neurons ensure the asymmetrical recruitment of ultra-fast primary motoneurons in the spinal cord. Neurons in this brainstem circuit, which encode rapid steering, receive salient visual inputs conveyed by specific V2a neurons in the pretectum. In contrast, the slow exploratory turn circuit is confined to two distinct brainstem nuclei, comprising eight pairs of V2a neurons.

Thus, our results show that separate sets of V2a neuron brainstem circuits govern rapid steering and exploratory turns. This modular organization adds a previously unknown dimension to the organization of brainstem circuits controlling locomotion, by revealing two distinct turning circuit modules and how their commands are integrated in the spinal circuits to encode the directionality of locomotion.

## Results

### Features of heading direction changes during swimming and steering

All modes of locomotion have the inherent ability to adjust their directionality through turns to avoid potential danger or pursue a target. To characterize the features of these turning movements, we used a high-speed camera to track swimming and turning in freely behaving larval zebrafish in response to optic flow (Fig. 1a, b). Two distinct turn patterns randomly occurred and could be distinguished. The first pattern was characterized by large asymmetrical changes in head and tail movement angles occurring over a short time that corresponds to rapid steering turns (Fig. 1c). The second pattern was marked by small changes in head direction while maintaining relatively symmetrical tail movements, which was typically associated with exploration during slow swimming[11,12]. We therefore referred to this as slow exploratory turn (Fig. 1c). Thus, zebrafish can flexibly adjust their head direction in response to asymmetrical optic flow by performing both fast steering and slow exploratory turns.

To further determine the characteristics of these two turning patterns, we first used an unsupervised algorithm (Time Series k-means, see "Methods") (Fig. 1d). Steering turns were characterized by large tail angles often exceeding 40°, while those of exploratory turns were always below 20° (Fig. 1d). The change in direction during both rapid steering turns and slow exploratory turns was initiated by asymmetrical tail bending, followed by changes in head direction angle (Fig. 1e, f). Rapid steering turns were characterized by large changes in both tail and head direction angles, which could be distinguished from those occurring during slow exploratory turns (Fig. 1f). Furthermore, there was a strong correlation between the tail angle and the amplitude changes in head direction (head angle) during rapid steering and slow exploratory turns (Fig. 1g).

To assess the brainstem circuits controlling the different turning patterns, we developed an in vivo head-fixed preparation with the ventral side up to further characterize turning movements (Fig. 1h). This preparation allows direct access and monitoring of the activity of ventrally located neurons during turning. Rapid steering-related turns were induced by the asymmetric optic flow, manifested as unidirectional large changes in tail bending (Fig. 1i). The supervised machine learning analysis revealed that these steering-related rapid turns were distinguished from swim-related exploratory turns, which were characterized by low-amplitude tail bending (Fig. 1j, k). The two turn patterns and their parameters in the ventral side-up preparation were similar to those induced in the dorsal side-up preparation (Supplementary Fig. 1a).

These results reveal two distinct turn patterns corresponding to rapid trajectory changes associated with fast steering and shallow routine turns associated with slow exploratory swimming. These two turn patterns are characterized by distinct kinematic features and directional changes, which are determined by asymmetrical changes in tail movements observed in both freely moving animals and those in head-fixed conditions.

### A distributed brainstem circuit encodes rapid heading direction change during steering

To determine the nature of the brainstem circuits encoding direction during rapid steering and slow exploratory turns, we used multiplane two-photon calcium imaging in head-fixed ventral side up preparation (Supplementary Fig. 1b). Brainstem neurons projecting to the spinal cord were identified following their retrograde labeling with a fluorescent dye injected into the spinal cord in the transgenic line Tg(elavl3:H2B-GCaMP6s) (Supplementary Fig. 1b). Rapid steering turns or slow exploratory turns were elicited by asymmetrical optic flow to the left or right, resulting in the activation of distinct neurons located in specific brainstem nuclei (Fig. 2a–f).

Rapid steering turns were encoded by a sparse circuit of projection neurons distributed across five brainstem nuclei (1 neuron in RoM1r; 1 neuron in RoV3; 2 neurons in the rostral part of MiV1; 1 neuron in the rostral part of MiV2; and 1 neuron in a caudal nucleus CaD) (Fig. 2a–c and Supplementary Figs. 1c and 2). Conversely, slow exploratory turns were encoded by confined circuits restricted to two nuclei, MiV1 and MiV2 (Fig. 2d–f and Supplementary Fig. 2). Remarkably, the brainstem neurons encoding slow exploratory turns were distinct from those encoding rapid steering turns. They were located more in the caudal parts of MiV1 and MiV2 nuclei. These results show that the brainstem circuit for exploratory turns displayed non-overlapping activity and distribution compared to those encoding rapid steering turns (Fig. 2d–f).

### Molecular identity of circuit neurons for turning

In mice, brainstem neurons encoding turning correspond to V2a neurons characterized by the expression of the transcription factor Chx10[24]. Therefore, we sought to determine whether the neurons comprising the circuits for rapid steering and slow exploratory turns in larval zebrafish share molecular identity with those in mice. We performed retrograde labeling of brainstem neurons in the transgenic line Tg(chx10:GFP)[27] and Tg(glyt2:GFP)[28]. The projection neurons in MiV1, MiV2 and RoV3 nuclei encoding rapid steering or slow exploratory turns all expressed the transcription factor Chx10 and correspond to excitatory V2a neurons (Supplementary Fig. 3a, b), while the neuron in the caudal nucleus, CaD, encoding only fast steering turns, expressed the transcription factor Dbx1 and correspond to glycinergic commissural V0d neurons. The molecular identity of CaD neurons was further confirmed using patch-seq single-cell RNA sequencing[29,30] (Supplementary Fig. 3c, d). These neurons express markers for glycinergic transmission and commissural V0d neuron identity.

These results show that two separate brainstem circuits encode direction changes during rapid steering and slow exploratory turns. The former is sparse and distributed across multiple brainstem nuclei, comprising mainly four pairs of V2a neurons and one pair of commissural inhibitory V0d neurons, while the latter is encoded by circuits confined to fewer nuclei comprising eight pairs of V2a neurons.

### A tuned recruitment of brainstem circuit neurons encoding rapid steering turns

While calcium imaging informs on the location and activity of neurons encoding rapid steering and slow explorative turns, it lacks the temporal resolution needed to determine the timing of the recruitment of individual neurons in the circuit. Therefore, we next used whole-cell patch-clamp recordings to more precisely determine the temporal

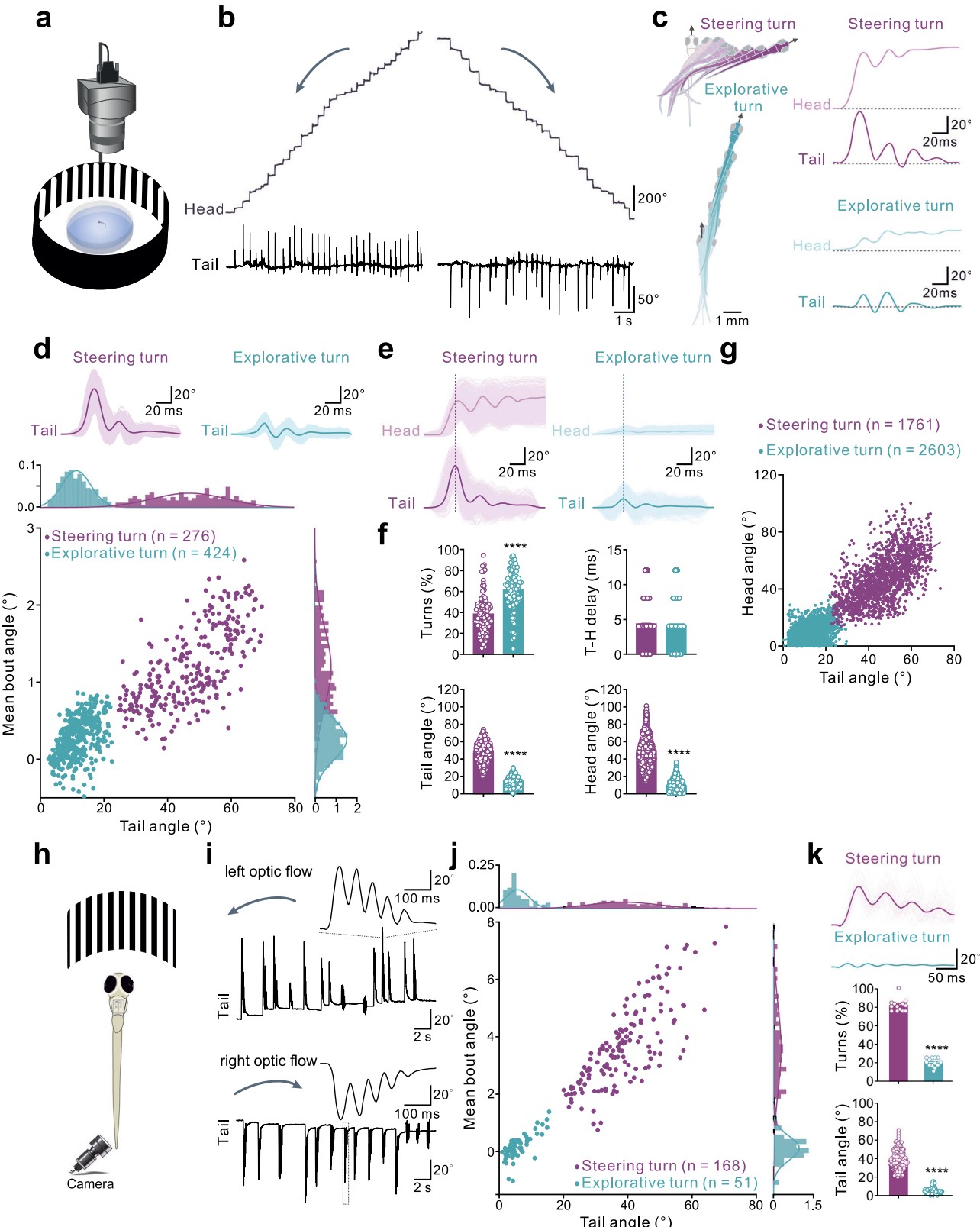

relationship between the firing of brainstem circuit neurons and the angle of the turn movements (Fig. 3 and Supplementary Fig. 4). Consistent with the calcium imaging data, the two pairs of rostral V2a neurons in MiV1, one pair of rostral V2a neurons in MiV2, and the V0d neurons in CaD, were exclusively recruited during rapid steering turns, while they remained inactive during slow exploratory turns (Supplementary Fig. 4a–j). In contrast, the caudally located V2a neurons in

both MiV1 and MiV2 nuclei were recruited only during slow exploratory turns, but not during rapid steering turns (Supplementary Fig. 4a–j).

We further revealed a strong relationship between the recruitment of V2a and V0d neurons encoding steering turns and the onset of rapid, large changes in tail angle (Fig. 3a–f). The recruitment of these steering-related V2a and V0d neurons was synchronized through

**Fig. 1 | Classification of rapid steering and slow exploratory turns. a** Setup with high-speed camera (250 fps) used to track optomotor response (OMR) induced by asymmetric optic flow. **b** Traces show cumulative heading direction change (upper) and tail bending amplitude (lower) during a 10-s OMR recording. **c** Left: Movement trajectories for steering turns (plum, upper) and exploratory turns (blue, lower). Right: Example traces of heading direction change and tail bending angle. **d** Locomotor events classified via unsupervised Time Series K-means (TSKmeans) with Dynamic Time Warping (DTW) and supervised cubic k-nearest neighbors (KNN). Features: mean bout angle and maximum tail amplitude. Classifier differentiates steering turns (plum, upper left; 276 events, 7 fish) and exploratory turns (blue, upper right; 424 events, 7 fish). Probability density displayed; each circle represents an event. **e** Larger dataset categorized using (**d**). Dashed lines show peak tail bending amplitude preceding heading change in both turn types. **f** Left upper: Turn occurrence (Paired t-test; $n = 92$ trials, 12 fish). Right upper: Time delay between peak tail bending and heading change (Mann–Whitney test; 92 events, 12 fish). Lower panels: Tail (left) and head angles (right) (Paired $t$ test; steering turns:

$n = 1761$, explorative turns: $n = 2603$). Circles represent individual events/trials. **g** Linear regression: significant correlation between tail bending amplitude and head direction change ($R^2 = 0.3916$ for steering, 1761 events; $R^2 = 0.1163$ for explorative, 2603 events; $N = 12$ fish). **h** Schematic of in vivo ventral side-up preparation with asymmetrical optic flow inducing turn movements recorded by a high-speed camera. **i** Turns generated in in vivo ventral side-up preparation. Dashed lines: example traces of left and right steering turns. **j** Classifier (as in **d**) categorized turn movements into steering (plum, 150 events, 6 fish) and explorative (blue, 68 events, 6 fish). Circles denote events. **k** Example tail angle traces within a single preparation (top; 3 trials, 1 fish; 58 events: 42 steering, 16 explorative). Middle: Turn occurrence (Mann–Whitney test; 13 trials, 6 preparations). Bottom: Tail angle for steering and explorative turns (Mann–Whitney test; 150 steering, 68 explorative; $N = 6$ fish). Plum = steering, blue = explorative. All data: mean ± SEM. Two-tailed $t$ test: ****$P < 0.0001$, ns not significant. Each circle = individual trial.

---

bidirectional electrical synapses mediated by gap junctions, ensuring their simultaneous firing (Fig. 3c, f). This electrical coupling allows for their concurrent recruitment as a circuit during steering turns to enable rapid changes in direction. These steering turn neurons were activated only when the change in tail angle exceeded ~40° (Fig. 3g–j) and subsequently deactivated when tail angles fell below 40° (Fig. 3b, e). In addition, there was a strong correlation between the number of action potentials (Fig. 3k, m) or the amplitude of the calcium response (Supplementary Fig. 4k) generated by neurons in the steering turn circuit and the turn angles, as indicated by changes in tail bending. However, no correlation was observed between the number of action potentials in these neurons and the tail beat frequency during rapid steering turns (Fig. 3l, n).

## Necessity of the sparse, distributed brainstem circuit for rapid steering turns

The results described above show that a sparse, distributed brainstem circuit comprising excitatory and inhibitory neurons encodes rapid changes in direction during steering turns. To determine if this circuit is necessary for generating rapid steering turns, we ablated either the steering V2a and V0d neurons individually or collectively and then tested the behavioral performance in the ablated fish using asymmetrical visual flow (Fig. 4 and Supplementary Fig. 5). Unilateral ablation of steering V2a neurons in MiV1 and MiV2 nuclei selectively impaired rapid steering turns towards the side of the ablation but not towards the contralateral side (Fig. 4a–d and Supplementary Fig. 5a–d). A similar effect was observed following unilateral ablation of only the inhibitory V0d neuron (Fig. 4e–h and Supplementary Fig. 5e–h). Furthermore, simultaneous ablation of both the steering V2a and V0d neurons resulted in a more pronounced impairment of rapid steering turns, as animals were no longer able to produce large changes in heading directionality (Fig. 4i–l and Supplementary Fig. 5i–l). The population data showed that ablation of both excitatory V2a and inhibitory V0d neurons impaired rapid steering turns but did not affect slow exploratory turns (Fig. 4k, l and Supplementary Fig. 5i–l). These results show that a sparse and distributed brainstem circuit essential for encoding rapid steering turns and for achieving large angle changes in direction.

## Brainstem steering circuit connects to spinal motoneurons to control rapid change in head direction

To determine how the combined activation of excitatory and inhibitory neurons in the steering brainstem circuit generates rapid asymmetrical direction changes, we performed paired whole-cell recordings from the constituent neurons. Reconstructions of the morphology of excitatory steering V2a neurons showed that these neurons project ipsilaterally along the entire length of the spinal cord (Fig. 5a). In contrast, the inhibitory V0d neuron displayed axons that

cross to the contralateral side of the brainstem before projecting along the entire contralateral spinal cord (Fig. 5e). The axons of both the excitatory and inhibitory neurons exhibited extensive collateral projections within each spinal segment (Fig. 5a, e). Paired whole-cell recordings from a steering V2a neuron and a spinal fast primary motoneuron revealed monosynaptic connections in the form of excitatory postsynaptic potentials (EPSPs) mediated via activation of ionotropic glutamate receptors (Fig. 5b, c). Activation of excitatory steering V2a neurons increased the firing frequency of ipsilateral primary motoneurons (Fig. 5d). In contrast, the inhibitory steering V0d neuron made monosynaptic connections with contralateral primary motoneurons, resulting in large inhibitory postsynaptic potentials (IPSPs) mediated by activation of glycinergic receptors (Fig. 5f, g). The activation of the inhibitory steering V0d neuron completely blocked the firing of contralateral fast primary motoneurons (Fig. 5h). The average synaptic delay of the connections between steering brainstem V2a neurons and the ultra-fast primary motoneurons was 0.8 ms, which is typical for monosynaptic connections. Thus, we reveal a brainstem-spinal circuit organization that controls steering turns underlying rapid directional changes. The simultaneous recruitment of excitatory and inhibitory steering neurons in the brainstem ensures an asymmetrical body bend by activating ipsilateral primary motor neurons while simultaneously inhibiting contralateral their counterparts, thereby enabling rapid steering turns.

In contrast to the steering V2a neurons, the V2a neurons located in the MiV1 and MiV2 nuclei, which are responsible for slow exploratory turns, make excitatory synaptic connections exclusively with secondary motoneurons and not with primary motoneurons (Supplementary Fig. 6a–j). Laser ablation of the MiV1 and MiV2 V2a neurons that are active during slow turns significantly reduced the occurrence of exploratory turns while increasing the frequency of rapid steering turns (Supplementary Fig. 6k–o). This finding suggests the existence of a distinct brainstem-spinal cord circuit dedicated to slow exploratory turns, which operates in parallel with the circuit responsible for rapid steering maneuvers.

## Pretectal V2a neurons control the brainstem steering circuit

Visual input plays an important role in controlling heading directionality[31–35]. To determine the brain regions and the identity of neurons encoding visual information and driving brainstem steering circuits, we examined the calcium dynamics in neurons of three key visual processing regions: the pretectum, optic tectum and thalamus using multiplane two-photon calcium imaging in the Tg(*elavl3:H2B-GCaMP6s*) fish line (Supplementary Fig. 7a). Neurons in the pretectum uniquely displayed direction selectivity in response to asymmetrical optic flow (Supplementary Fig. 7b), while neurons in the optic tectum and thalamus were activated without any direction selectivity

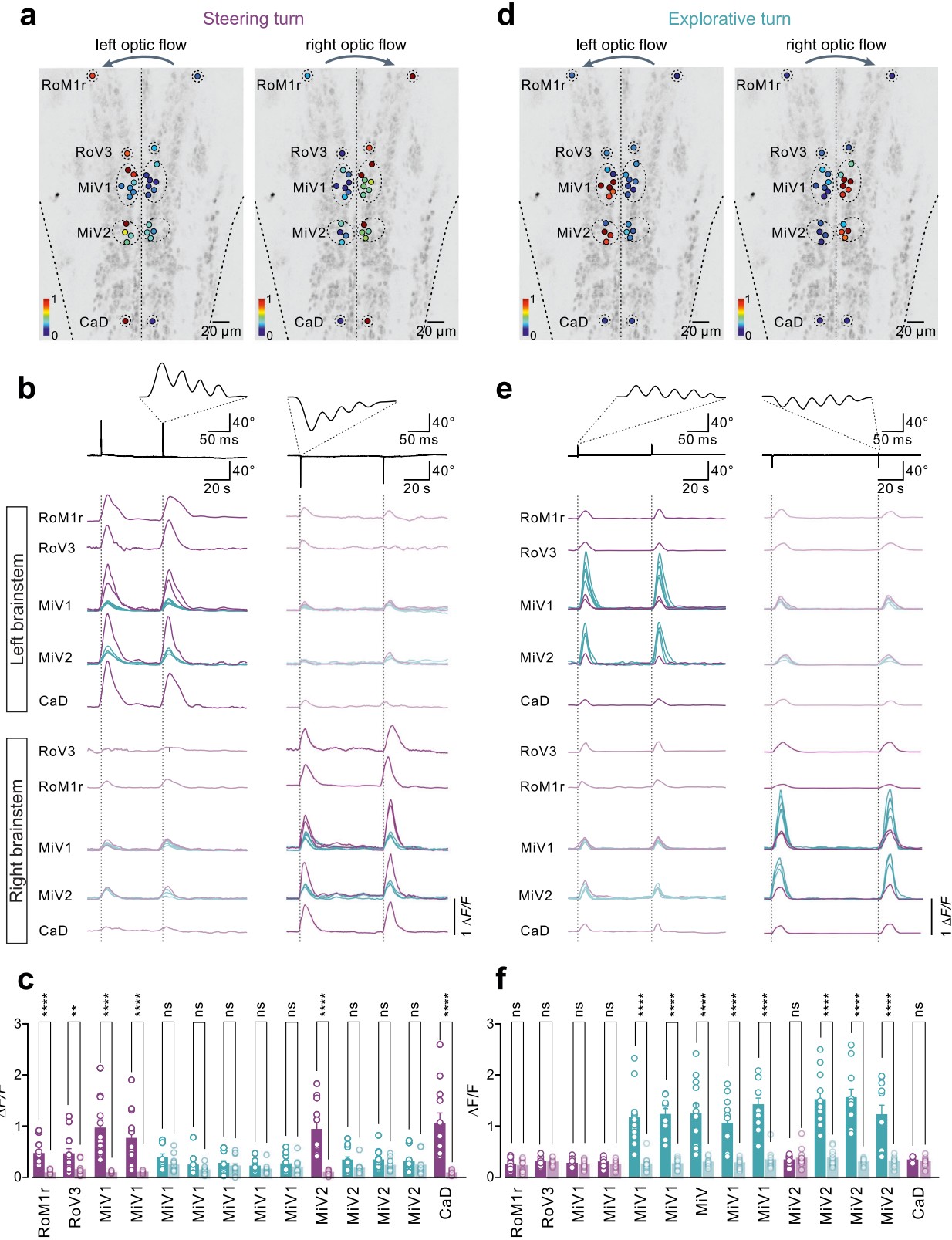

(Supplementary Fig. 7b, c). This is consistent with previous studies showing that direction-selective neurons in the pretectum integrate visual information from asymmetrical optic flow and are closely correlated with the optomotor response[36–39]. We further revealed that the direction selective pretectal neurons encode steering turns, not exploratory turns, underlying direction changes (Supplementary Fig. 8a–d). For steering turns, there was significantly asymmetrical

activation of pretectal neurons with ipsilateral neurons being significantly more excited than the contralateral ones in terms of both the number of recruited neurons and the calcium change intensity (Supplementary Fig. 8b, c). There was a linear relationship between the amplitude of tail bending movement and activity intensity or the number of recruited neurons in the ipsilateral but not the contralateral pretectum (Supplementary Fig. 8d). However, during exploratory

**Fig. 2 | Distinct brainstem circuits encode steering turn and exploratory turn.** **a**–**c** Graph showing soma distribution and activation intensity of spinal projecting neurons (SPNs) in distinct nuclei of the brainstem during rapid steering turns induced by left or right optic flow (**a**). Excitatory neurons were distributed across four brainstem nuclei (2 neurons in the rostral part of MiV1; 1 neuron in the rostral part of MiV2; 1 neuron in RoM1r, and 1 neuron in RoV3) and an inhibitory neuron (1 neuron in CaD). These neurons displayed significantly larger calcium responses during ipsilateral rapid steering turns, while the rest of the neurons showed weak or no response (**b**, **c**) ($N = 12$ fish; paired $t$ test). **d**–**f** Graph showing soma distribution and activation intensity of SPNs in distinct nuclei of the brainstem during explorative turns induced by left or right optic flow (**d**). Excitatory neurons

distributed across the caudal part of both MiV1 and MiV2 displayed significantly larger calcium responses during ipsilateral explorative turns, while the rest of the neurons showed weak or no response (**e**, **f**) ($N = 12$ fish; paired $t$ test). The color saturation of each point indicates the normalized maximum $\Delta F/F$ of this neuron (**a**, **d**). The color plum represents neurons that are active during steering turns, while blue indicates neurons that are active during explorative turns. The solid color depicts neurons that are ipsilateral to the direction of the turn, whereas the faded color represents neurons that are contralateral to the turn movements. Each circle corresponds to a single neuron. All data are presented as mean ± SEM. The statistical test used was two-tailed $t$ test. ****$P < 0.0001$, $P = 0.0053$ for RoV3, ns denoting no significant difference.

turns, neurons in both the left and right hemispheres of the pretectum were simultaneously activated by asymmetrical visual flow (Supplementary Fig. 8b, c).

We also found anatomical evidence that neurons of the brainstem steering circuit receive direct axon innervation from pretectal neurons. We used the freely available dataset of single neuron labeling (https://mapzebrain.org/atlas/3d) to analyze the axon-projecting regions of direction-selective pretectal projection neurons (PPNs). The majority of these neurons projected to the brainstem and sent their axons to the three nuclei, MiV1, MiV2 and CaD, forming a circuit for turning (Supplementary Fig. 9a). The density plot of soma locations revealed that these PPNs were predominantly located in the ventral pretectum (Supplementary Fig. 9a).

A key neuronal population of the ventral pretectum is V2a neurons (Fig. 6a), and therefore we examined if this population directs fast steering turn movements. We first examined the activity of pretectal V2a neurons during direction changes induced by asymmetrical visual flow using the Tg(Chx10:Gal4;UAS:GCaMP6s) fish line (Fig. 6a–d). During steering turns, there was asymmetric activation of pretectal V2a neurons in terms of both the number of recruited neurons and the calcium change intensity (Fig. 6c). The activation intensity and the number of recruited pretectal V2a neurons correlated linearly with the amplitude of turning movements (Fig. 6d). However, pretectal V2a neurons were bilaterally and symmetrically recruited during exploratory routine turns (Fig. 6b–d). Moreover, we investigated whether the brainstem steering circuit neurons were electrically coupled to the PPNs using dye coupling. Injection of a high concentration of neurobiotin through the patch pipette into the steering V2a and V0d neurons traced their dye-coupled V2a neurons distributed in the ventral pretectum (Supplementary Fig. 9b, c). The existence of electrical coupling between pretectal V2a neurons and steering V2a/V0d neurons was further confirmed using paired whole-cell recordings (Supplementary Fig. 9d). The induced EPSPs in the steering turn-related V2a neurons, triggered by pretectal V2a neurons, are mediated by mixed electrical and chemical synapses. The application of the gap junction blocker carbenoxolone reduced the electrical component of the EPSP (Supplementary Fig. 8e). These results indicate that pretectal V2a neurons are the main PPNs driving the neurons of the brainstem steering circuits via mixed electrical and chemical synaptic connections.

During steering turns, the ipsilateral pretectal V2a neurons exhibited significantly larger calcium responses compared to their contralateral counterparts, indicating that unilateral activation of pretectal V2a neurons induces ipsilateral steering turns (Supplementary Fig. 8). Next, we used the in vivo dorsal-up preparation (Fig. 6e–j) to assess the impact of unilateral activation of pretectal V2a neurons on brainstem steering circuit neurons and steering turn movements. We used a transgenic line, Tg(Chx10: Gal4; UAS: ChR2-mCherry), in which channelrhodopsin was selectively expressed in V2a neurons (Fig. 6e). Unilateral optogenetic stimulation of the ventral pretectum activated the recorded V2a neurons, whose firing rate increased in response to the intensity of light stimulation (Fig. 6f). Optogenetic activation of unilateral pretectal V2a neurons elicited a large excitatory

drive in the ipsilateral steering V2a and V0d neurons, which persisted in a high $Ca^{2+}/Mg^{2+}$ solution but was blocked by NBQX and APV (Fig. 6g, h). Furthermore, optogenetic activation of unilateral pretectal V2a neurons reliably generated steering turns but not exploratory turns (Fig. 6i, j). The amplitude of tail bending underlying steering turns increased with the intensity of the light stimulation (Fig. 6i, j). These results were further confirmed by electrical stimulation of the ventral region of the unilateral pretectum, which generated large excitation in the brainstem steering V2a and V0d neurons and reliably produced steering turns in an intensity-dependent manner (Supplementary Fig. 9f–i). Finally, we showed that unilateral ablation of pretectal V2a neurons impaired the ability of fish to generate ipsilateral steering turns underlying rapid direction changes but did not affect direction changes on the contralateral side of the ablation (Fig. 6k–m and Supplementary Fig. 10). The percentage of occurrence, tail angles and head angles for the ipsilateral rapid steering turns decreased significantly after laser ablation (Fig. 6m and Supplementary Fig. 10).

These results provide a causal relationship between the activity of pretectal V2a neurons and their role in transforming visual information into steering commands that are conveyed specifically to brainstem circuits controlling rapid direction changes. Thus, the pretectum-brainstem steering circuit serves as a sub-feature of visual system that directs movement trajectory.

## Discussion

Turning movements are essential for vertebrates to flexibly control their direction of movement. Sharp turns facilitate rapid trajectory changes, while shallow turns enable slow, exploratory behaviors[11,12]. In their exploration of the environment, zebrafish primarily exhibit slow swimming behavior, characterized by frequent directional changes through shallow, exploratory turns[11]. However, sudden changes in direction are produced by rapid and large steering turns[12]. Two circuit organizations can be envisioned for controlling these turning patterns. The first is a unified circuit that adjusts its activity to match the turn angle to the behavioral necessity for both rapid steering and routine exploratory turns. The second is a two-circuit module organization, where the two turning patterns are encoded by distinct circuits, forming two parallel descending pathways to generate either rapid steering turns or slow exploratory turns.

By combining behavioral analysis, electrophysiology, functional imaging, and optogenetics, we have resolved the circuit underpinnings for rapid steering turns and shallow exploratory turns. Our results first show that these two turning patterns can be behaviorally distinguished based on their kinematics. Rapid steering turns were characterized by large changes in tail angles, often exceeding 40°, while slow shallow turns were associated with small changes in tail angle, consistently remaining below 20°. These slow, shallow turns typically occur during slow exploratory swimming[11,12]. Furthermore, our results reveal a two-circuit modular organization in which the two turn patterns are encoded by distinct brainstem circuits. Rapid steering turns are encoded by a sparse brainstem circuit consisting of four pairs of V2a neurons and one pair of commissural V0d neurons, which are distributed across separate nuclei. These turns are generated by strong

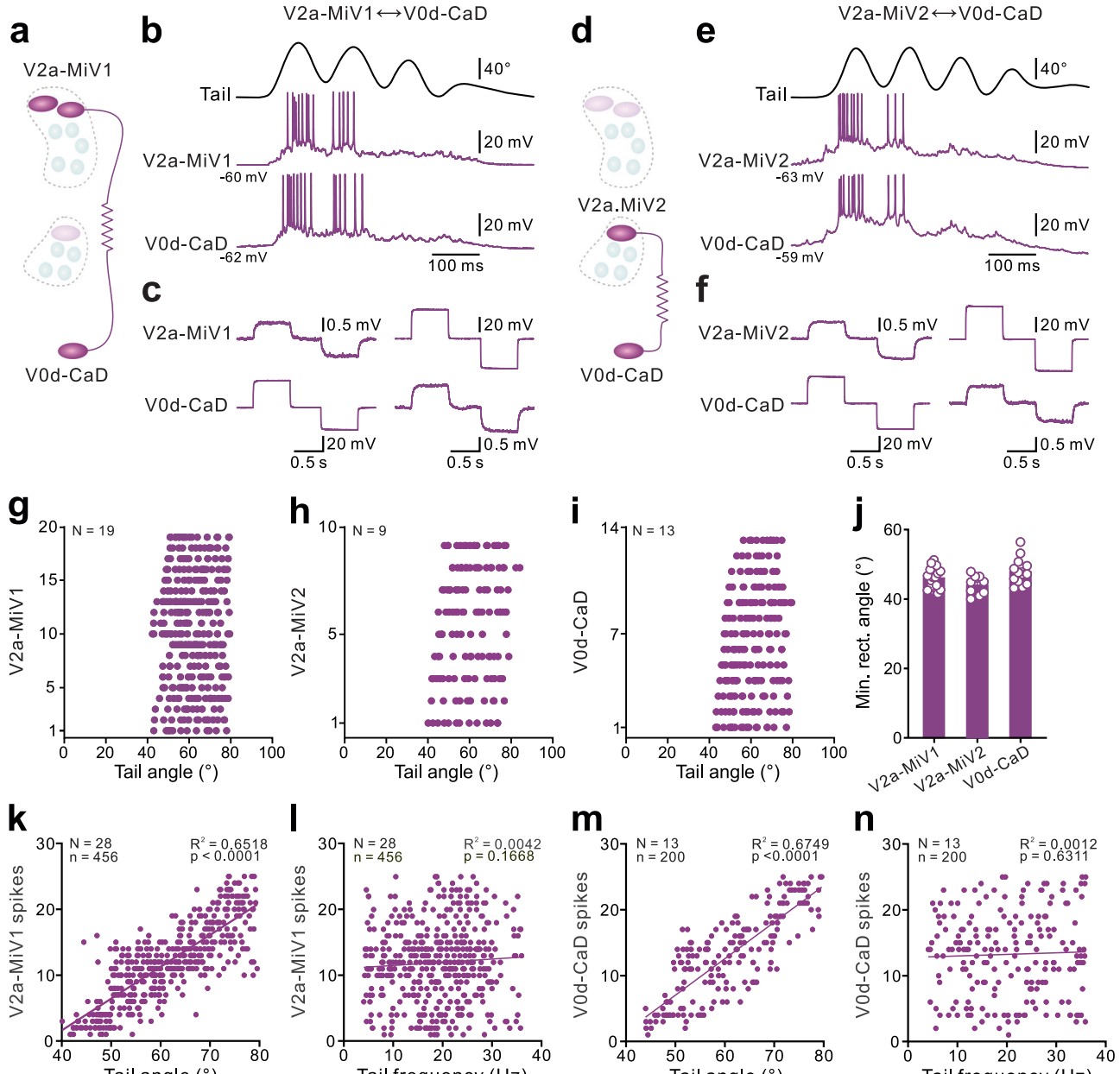

**Fig. 3 | The recruitment of steering neurons was correlated to tail-bending amplitude. a–c** Paired whole-cell patch-clamp recordings between a steering V2a neuron in the MiV1 and a steering V0d neuron in the CaD (**a**) reveal that the two neurons simultaneously discharged action potentials during an ipsilateral steering turn (**b**). The steering V2a and V0d neurons are electrically coupled through gap junctions (**c**). **d–f** Paired whole-cell patch-clamp recordings between a steering V2a neuron in MiV2 and a steering V0d neuron in CaD (**d**) reveal that the two neurons were simultaneously recruited during an ipsilateral steering turn (**e**). The steering V2a and V0d neurons are electrically coupled through gap junctions (**f**). **g–j** Graphs showing the recruitment tail angle for the steering V2a neurons in MiV1 ($N = 19$ neurons from 19 preparations, $n = 318$ events), MiV2 ($N = 9$ neurons from 9 preparations, $n = 138$ events), and the steering V0d neurons in CaD ($N = 13$ neurons from 13 preparations, $n = 200$ events) during a steering turn. Statistical graph showing the minimal recruitment angles for the above neurons in (**g–j**). **k–n** Graphs showing a linear relationship between tail angle and the number of action potentials generated in the recorded steering V2a neurons (**k**) and V0d neurons (**m**). No such relationship exists between tail beat frequency and the number of action potentials generated in the steering V2a neurons (**l**) and V0d neurons (**n**). A two-sided test was applied to determine the strength and direction of the correlation. The coefficient of determination ($R^2$) represents the proportion of variance explained by the model, and the $P$ value evaluates the statistical significance of the correlation. $P < 0.05$ was considered statistically significant.

activation of ipsilateral ultra-fast primary motoneurons by descending V2a neurons, accompanied by simultaneous inhibition of the corresponding contralateral motoneurons. Ablation of these neurons impaired the execution of rapid steering turns while leaving slow exploratory turns unaffected. In contrast, slow exploratory turns associated with slow swimming are encoded by eight pairs of V2a neurons confined to two distinct brainstem nuclei. These turns are generated by an asymmetrical imbalance of activity between the V2a

neurons on either side of the brainstem, without involving simultaneous inhibition. Ablation of these neurons impaired the execution of slow exploratory turns while not affecting rapid steering turns. Our results reveal a previously unknown modular organization of brainstem circuits that control both steering and exploratory turns, as well as the directionality of locomotion in vertebrates.

Locomotion initiation, speed control, termination, and directionality are orchestrated by descending brainstem neurons[1,3,5,7,13,23,40,41].

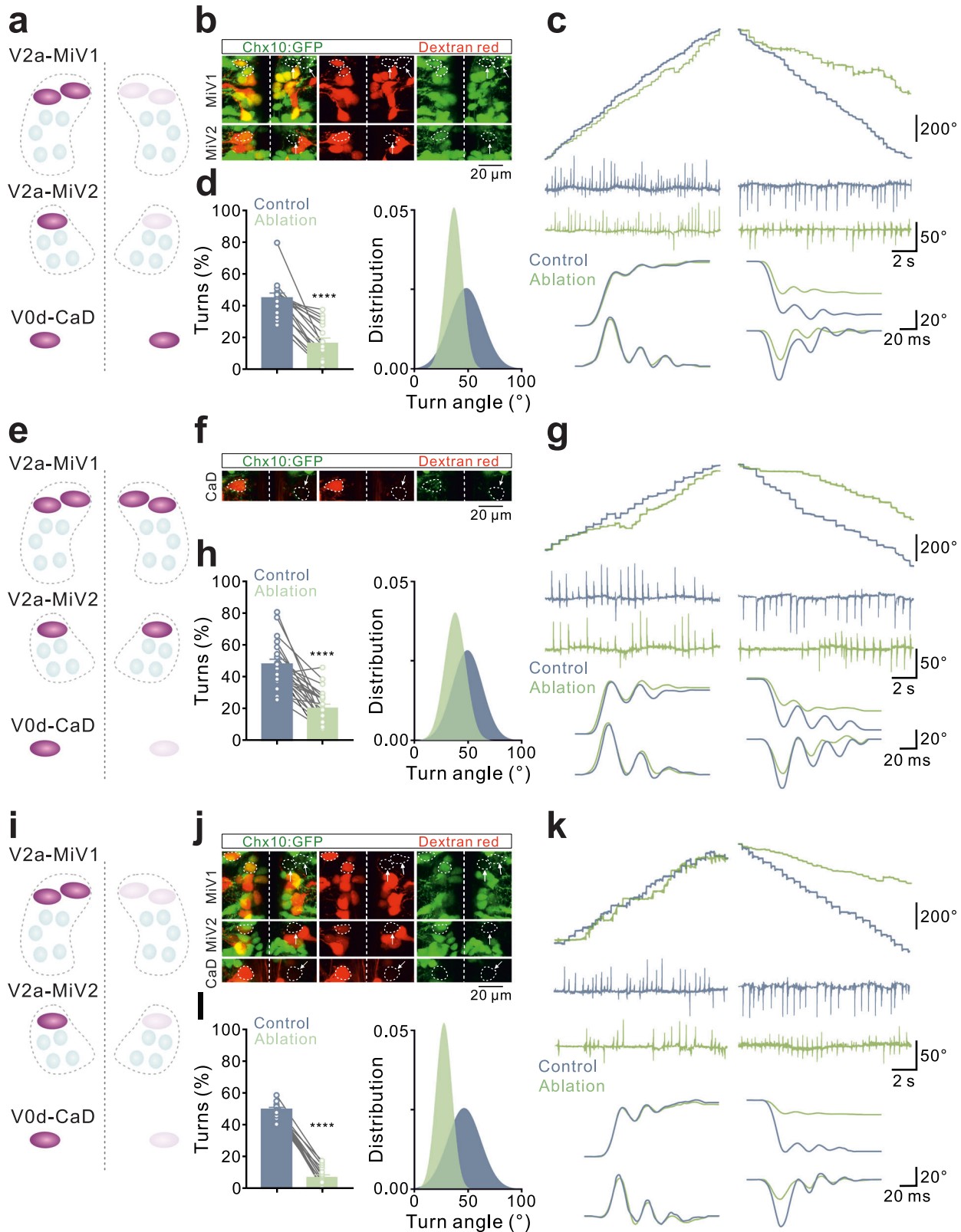

These neurons convey specific commands to spinal circuits to generate movements with precise timing, force, and trajectory[14,24,30,42–47]. However, understanding the organization and precise connections of these brainstem command neurons with spinal neurons has been challenging due to the complexity of brainstem nuclei. By leveraging a combination of techniques, we have uncovered the true organization of two brainstem circuits and their connections to spinal motoneurons for controlling different turn patterns. Our results show that inhibitory and excitatory steering brainstem neurons function as an ensemble, coupled by gap junctions, to generate sharp steering turns. The recruitment of these steering command neurons is correlated with the amplitude of direction change rather than locomotor frequency. Traditionally, it has been widely accepted that descending excitatory commands are fundamental in setting locomotor parameters[6].

**Fig. 4 | Ablation of brainstem steering neurons impaired steering turn.**
**a**–**d** Diagram showing unilateral ablation of the steering V2a neurons in the MiV1 and MiV2 nuclei (**a**). Image stacks showing unilateral ablation of the steering V2a neurons in the MiV1 and MiV2 nuclei as revealed by retrograde traceing in the Tg(*Chx10:GFP*) fish line (**b**). Unilateral ablation of the steering V2a neurons significantly reduced both the cumulative angle of heading direction change and the tail angle during ipsilateral steering turns, while leaving the contralateral side unaffected (**c**). Statistical analysis showing that, compared to the control group, there was a significantly decreased in the occurrence of steering turns and the distribution of of tail angle amplitude following the ablation of the steering V2a neurons (**d**) (Paired *t* test, *N* = 16 fish). **e**–**h** Diagram showing unilateral ablation of the steering V0d neuron in CaD (**e**). Image stacks showing the unilateral ablation of the steering V0d neuron in the CaD in the Tg(*Glyt2:GFP*) fish line (**f**). Unilateral ablation of the steering V0d neurons reduced the cumulative angle of heading

direction change and the tail angle during the ipsilateral steering turn, but did not affect the contralateral side (**g**). Statistical analysis showing a significantly decrease in occurrence of steering turns and the distribution of the tail angle amplitude following ablation of the steering V0d neurons (**h**) (Paired *t* test, *N* = 22 fish). **i**–**l** Diagram showing unilateral ablation of the steering V2a and V0d neurons (**i**). Image stacks showing the unilateral ablation of steering V2a and V0d neurons that is confirmed by retrograde tracing in the Tg(*Chx10:GFP*) fish line (**f**). Unilateral ablation of the steering V2a and V0d neurons resulted in a reduction in the heading direction angle and the amplitude of the tail angle of the ipsilateral steering turn, but did not affect the contralateral side (**g**). Statistical analysis showing a significantly decreased in the occurrence of steering turns and the distribution of the of tail angle amplitude following ablation of the steering V2a and V0d neurons (**h**) (Paired *t* test, *N* = 25 fish). All graphs show mean ± SEM. The statistical test used was two-tailed *t* test. \*\*\*\**P* < 0.0001, each circle representing a fish.

However, the role of inhibitory descending commands in locomotion has received less attention. Our findings reveal that inhibitory brainstem steering neurons project axons across the midline to inhibit contralateral spinal ultra-fast primary motoneurons. Their temporal burst firing can promptly halt the discharge of contralateral spinal motor neurons, facilitating relaxation of the contralateral axial muscle and promoting body bending. Moreover, excitatory brainstem steering neurons act synergistically with their inhibitory counterparts by directly exciting ipsilateral spinal motoneurons, inducing powerful contractions of axial muscles on the turn side. Laser ablation of both inhibitory and excitatory brainstem steering neurons significantly impaired rapid steering turns, without affecting shallow routine turns during swimming. This underscores the crucial role of inhibitory descending commands, which appear to be as essential as excitatory commands in steering locomotion.

In vertebrates, including mammals, the optic tectum (referred to as the superior colliculus in mammals) has been identified as the visual processing center that potentially drives brainstem nuclei to achieve locomotor asymmetry[32,35,48–58]. In lampreys, the brainstem-projecting neurons located in the deep layers of the tectum are responsible for selecting either an escape response triggered by fast looming stimuli or an orienting response induced by slow looming stimuli[48]. In zebrafish, distinct subpopulations of tectal projection neurons, revealed by optogenetic mapping and single-cell atlas of the tectum, encode escape and approach behaviors, respectively[33]. Optic flow provides animals with information about their own actions in relation to their surroundings, which is crucial for the proper control of heading direction of self-motion[59,60]. Consequently, the asymmetric optic flow stimuli can continuously induce changes in heading direction and turning movements in both lampreys and zebrafish[36,37,50,51,61,62]. A subset of neurons in the pretectum that serve as direction-translation neurons, receiving synaptic inputs from direction-selective retinal ganglion cells (RGCs) that detect the visual flow[33,34,36,37,50,51,61,62]. However, the identity of pretectal direction-selective neurons and their role in translating visual information into the required locomotor commands have remained elusive. Our results now show that V2a neurons in the ventral pretectum process the asymmetric visual flow. These neurons project axons ipsilaterally across the brainstem to form monosynaptic glutamatergic synapses and gap junctions with brainstem circuit neurons that encode steering turns. These pretectal V2a neurons are crucial for encoding the steering turns underlying rapid changes in direction. Optogenetic activation of these pretectal V2a neurons simultaneously recruits both excitatory and inhibitory brainstem circuit neurons, generating rapid steering turns. Increasing the intensity of electrical and optogenetic stimulation in a gradual manner mimicked the recruitment pattern of pretectal neurons, resulting in progressively greater amplitudes of head direction changes and tail bending. Conversely, ablation of these pretectal neurons impairs sharp steering turns without affecting slow exploratory turns. This is consistent with a previous study demonstrating that the pretectum is

crucial for visually-induced acute turns in lamprey[63]. Thus, our findings demonstrate that the neural circuits formed between the pretectum and brainstem faithfully transform direction-related visual information into steering turning movements. Our results show that the optic tectum and thalamus do not show any bias in their activity on the two-sides of the brain in response to asymmetric visual flow. These regions could process other types of visual information such as looming to produce turning movements.

Studies in mice have revealed that V2a neurons in the nucleus gigantocellularis (Gi) control locomotor asymmetry[5]. These neurons serve a dual function: bilateral activation halts locomotion[64], while unilateral stimulation biases locomotor direction and induces turning[24]. The turning pattern and stopping are encoded by different V2a-Gi neurons that target distinct parts of the spinal cord along the rostrocaudal axis[24,64,65]. The V2a-Gi neuron population is not uniform but consists of distinct subsets that are anatomically and functionally specialized for specific spinal targets. Cervical-projecting V2a-Gi neurons control head yaw rotation and subsequent changes in trajectory without causing locomotor arrest, while lumbar-projecting V2a-Gi neurons trigger a rapid locomotor arrest without head rotation or directional changes[65]. However, how the turning and stopping commands are decoded within the spinal circuits remains unclear. It has been shown that brainstem-descending V2a neurons are pivotal in the postnatal development and refinement of locomotor behaviors[66]. The involvement of brainstem V2a neurons in turning behavior in zebrafish was previously reported by Huang et al.[26], who found that ablation of specific neurons affected turning behavior. Our study adds missing aspects to the understanding of locomotor direction control. First, turning behavior can be deconstructed into rapid steering associated with sharp changes in direction and shallow routine turns associated with exploratory swimming. Second, these two turning patterns are encoded by distinct brainstem circuits involving separate V2a neurons, even when located within the same nucleus. Third, steering turns require not only ipsilateral excitation of ultra-fast motoneurons in the spinal cord but also simultaneous descending inhibition of contralateral motoneurons. In contrast, slow exploratory turns are encoded merely by asymmetrical activation of descending V2a neurons. Thus, our results show that the control of turning angles is not simply a matter of increasing activation of a single brainstem circuit but relies on two distinct circuits, each computing turning in a specific way. There is a significant degree of conservation in the molecular identity of the neuronal types encoding turning between fish and mammals. Therefore, it is likely that separate parallel brainstem circuits exist, in addition to V2a-Gi neurons, to control steering, as revealed in our study.

## Methods
### Animals
Zebrafish (*Danio rerio*) were raised and maintained using an automatic fish-housing system in a zebrafish facility at Tongji University as

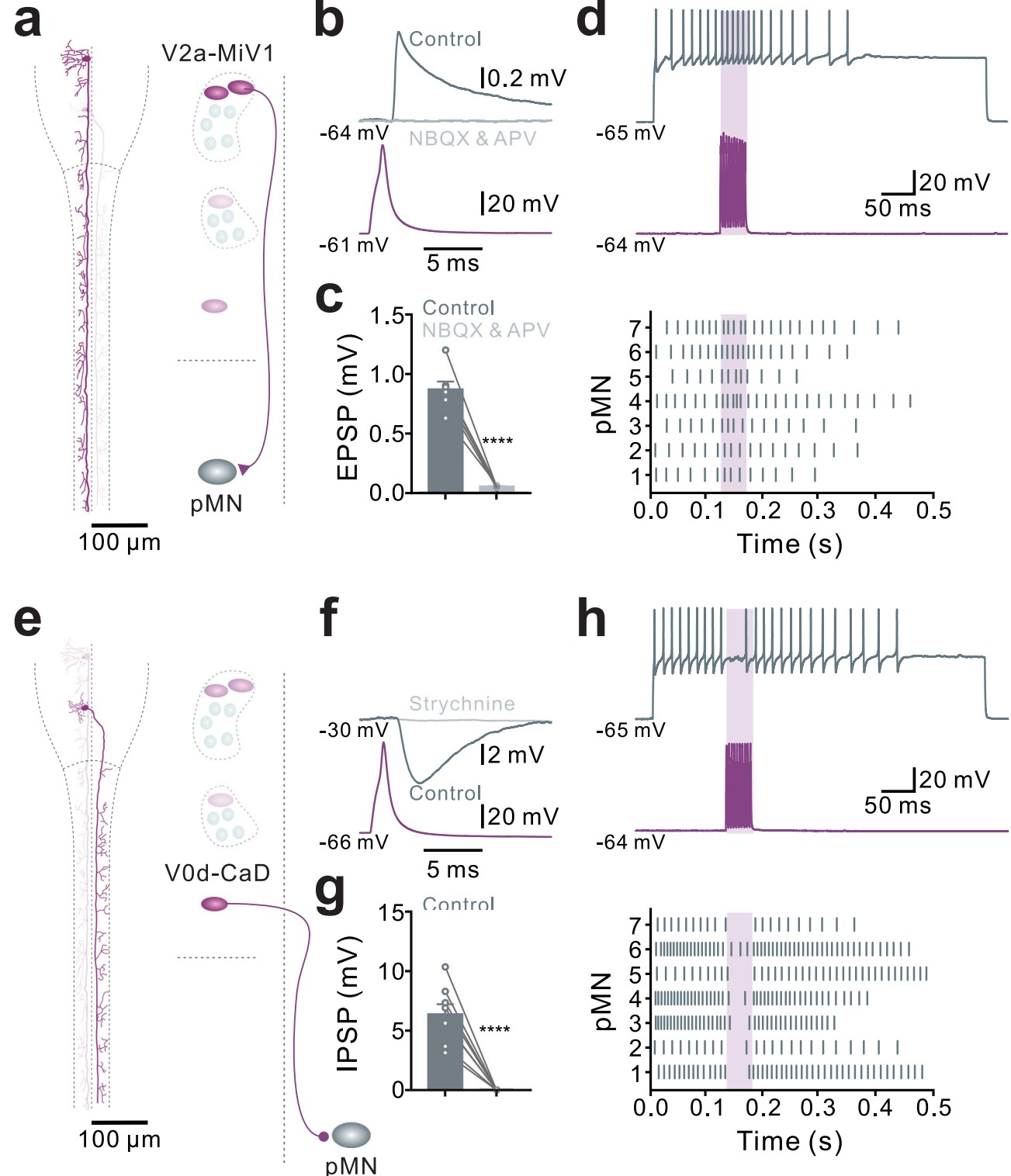

**Fig. 5 | The steering V2a neurons project ipsilaterally to excite spinal moto-neurons, while the steering V0d neurons project contralaterally to inhibit spinal motoneurons. a–d** Reconstructed morphology of a steering V2a neuron projecting ipsilaterally along the entire length of the spinal cord (**a**). An action potential generated in the steering V2a neurons triggered EPSPs in the spinal primary motoneurons, which were blocked by the application of NBQX and AP5 (**b**; grey trace). The population data for **b** (**c**; Paired *t* test, *n* = 7 pairs from 7 preparations). The induced high-frequency firing in the steering V2a neurons increased the firing frequency of the discharging postsynaptic primary motoneurons (**d**; *N* = 7 pairs from 7 preparations). **e–h** Reconstructed morphology of a steering V0d

neuron shows the axon crossing the midline at the brainstem and projecting con-tralaterally along the entire length of the spinal cord (**e**). An action potential gen-erated in the steering V0d neurons triggered IPSPs in the recorded spinal primary motoneurons, which were blocked by the application of strychnine (**f**; grey trace). The population data for (**f**) (**g**; Paired *t* test, *n* = 8 pairs from 8 preparations). The induced high-frequency firing in the steering V0d neurons inhibit the firing of the postsynaptic primary motoneurons (**h**; *N* = 7 pairs from 7 preparations). All data are presented as mean ± SEM. The statistical test used was two-tailed *t* test. ***P < 0.001, ****P < 0.0001. Each circle represents a single neuron.

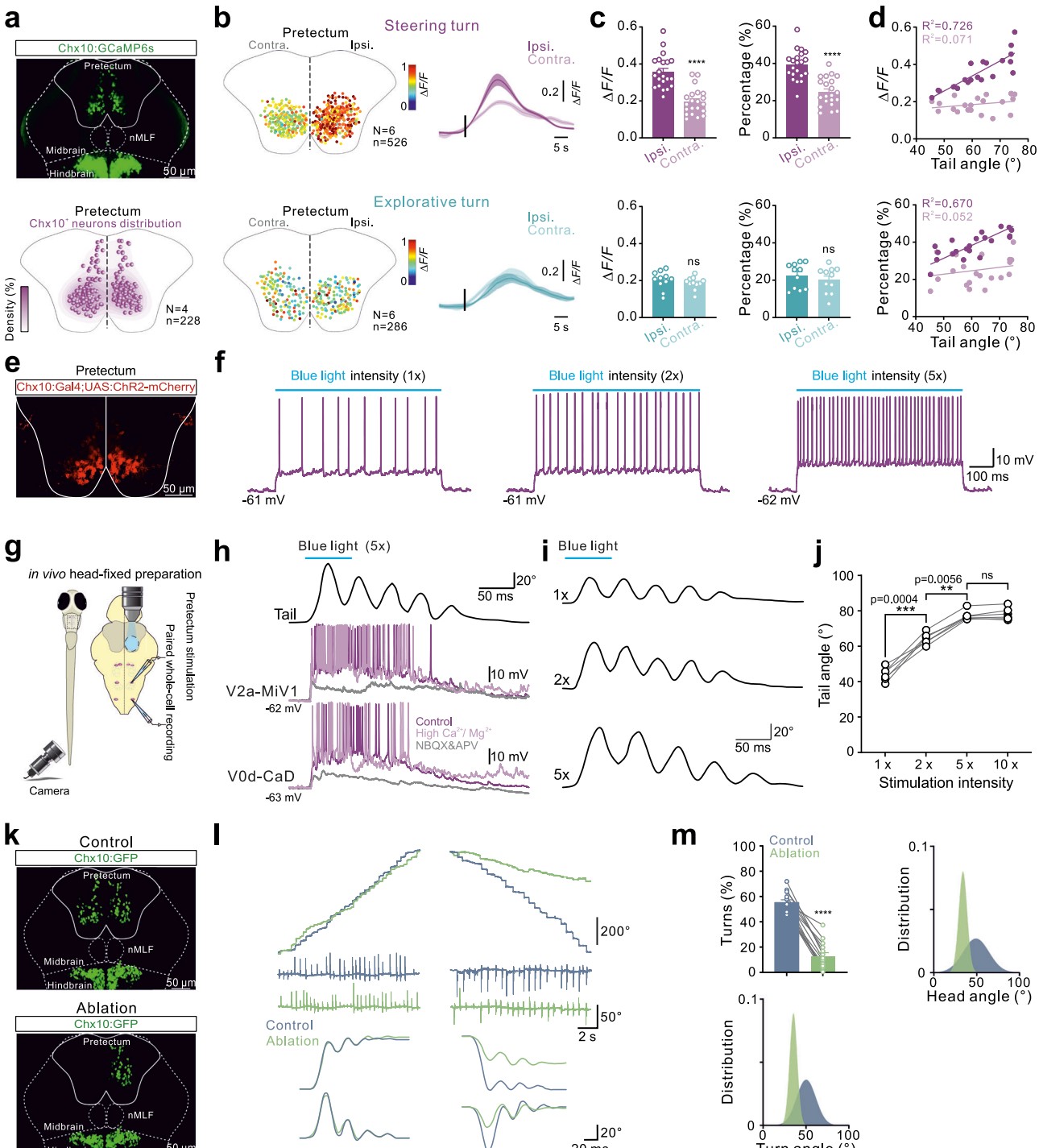

**Fig. 6 | Pretectal V2a neurons drive brainstem steering circuit and generate steering turn movement. a** Upper: Soma distribution of V2a neurons in *Tg(Chx10:GCaMP6s)*. Lower: Statistical analysis of V2a soma distribution in the pretectum. **b, c**, Statistical analysis of soma distribution and response strength of recruited pretectal V2a neurons during steering (left upper; *n* = 526, *N* = 6) and explorative turns (left lower; *n* = 286, *N* = 6). Right upper: Ipsilateral pretectal V2a response significantly larger than contralateral during steering. Right lower: No significant difference during explorative turns. Population data (Unpaired *t* test, 21 trials from 6 fish for steering, 11 trials from 6 fish for explorative; circles = individual trials). **d** Calcium response (ΔF/F) and percentage of recruited ipsilateral V2a neurons showed a linear relationship with increasing tail bending amplitude during steering (21 trials, 6 fish; circles = individual trials). **e, f** ChR2-mCherry expression in pretectal V2a neurons. Optogenetic activation in *Tg(Chx10:Gal4; UAS:ChR2-*

*mCherry)* in a light-intensity-dependent manner. **g–j** Experimental setup for optogenetic stimulation and high-speed recording (**g**). Stimulation of pretectal V2a neurons activated steering V2a and V0d neurons, inducing a steering turn. Neuronal firing persisted in high Ca²⁺/Mg²⁺ but was blocked by NBQX/AP5 (**h**). Tail angle amplitude increased with blue light intensity until plateau (**i, j**; one-way ANOVA, *N* = 6 fish; circles = individual preparations). **k–m** Stacked images of pretectal V2a neuron ablation in *Tg(Chx10:GFP*; **k**). Unilateral ablation reduced cumulative heading direction changes and tail angle during ipsilateral steering but not contralateral turns (**l**). Steering turn occurrence significantly decreased postablation (Paired *t* test, *N* = 14 fish). Fish failed to generate ipsilateral steering turns, producing only swim-related turns after ablation (**m**; *N* = 14 fish; circles = individual preparations). All data: mean ± SEM. Two-tailed *t* tests: ****P < 0.0001, ***P < 0.001, **P < 0.01, ns not significant.

described previously[16]. Both wild-type and several transgenic zebrafish lines were utilized in this study, including Tg(*Glyt2:GFP*)[28], Tg(*Chx10:GFP*)[27], Tg(*elavl3:H2B-GCaMP6s*)[40], Tg(*Tol-O56:GFP*)[67] and Tg(*Chx10:Gal4;UAS:ChR2-mCherry*)[68]. Zebrafish larvae, aged 12-14 days post-fertilization (dpf), were used for experiments that were conducted at room temperature (24–26 °C). All experimental protocols were approved by the Animal Use Committee of Tongji University and performed in accordance with laws and regulations in China.

## Behavioral recording in free-swimming zebrafish

Free-swimming larval zebrafish were placed in a Petri dish within a soundproof behavioral box, where they acclimated for 20 min before the experiments. Illumination was provided under the dish using a white LED panel, and the zebrafish were exposed to asymmetrical grating drift as visual stimuli (1 cm stripe, rotating at 0.9 rad/s). The Petri dish with a radius of 4 cm was positioned 2 cm away from the grating apparatus. Behavioral responses were captured by a high-speed camera (MC1362, Mikrotron) equipped with a lens (HC3505A, CHIOPT) and operating at 250 fps. The camera was connected to a DVR Express Core 2 acquisition system (IO Industry) and controlled via CoreView software. Each experiment consisted of four 25-s trials, two with clockwise and two with counterclockwise visual stimulation. Each trial was separated by a 10-min interval to prevent habituation.

## In vivo preparation

An in vivo preparation with intact eyes was established for detecting the activity of single neurons during turning movement using either whole-cell patch-clamp recording or two-photon imaging. To visualize the neurons of the MiV and CaD nuclei, animals were anesthetized with 0.03% tricaine methane sulfonate (MS-222, Sigma-Aldrich) solution. Subsequently, dextran tetramethylrhodamine 3000 MW (D3307, Thermo Fisher) was injected into the 16–17th segments of the spinal cord using dye-soaked pins. The fish with tracer injection was left overnight in darkness to ensure effective retrograde transport. Subsequently, the fish were deeply anesthetized in a HEPES solution containing 0.03% MS-222 and eviscerated. The dorsal-up or ventral-up positioned fish were immobilized with tungsten needles at the mouth and otic regions. Using tungsten needles, the skull at the hindbrain was carefully opened, and the bone were gently removed with forceps to fully expose the dorsal area of the hindbrain. A horizontal incision was then made along one otic region underneath the brain, extending to the opposite side. The skull or the bones wrapping the hindbrain was removed to completely expose the entire hindbrain. The dissection was carefully and gently performed to avoid compressing or damaging the brain tissue while retaining the intact eyes and tail. The preparation was continuously perfused with the HEPES solution (in mM: 134 NaCl, 2.9 KCl, 2.1 CaCl$_2$, 1.2 MgCl$_2$, 10 HEPES, 10 glucose, pH 7.8 adjusted with NaOH, osmolality 290 mOsm). No pharmacological agents were used to block neuromuscular junctions, ensuring free tail movements during experiments.

For the in vivo preparation for double whole-cell patch-clamp recordings, we used the Tg(*Tol-O56:GFP*) fish line, which expresses GFP in spinal primary motoneurons. The ultra-fast motor neurons correspond to primary motor neurons identified by GFP expression and their large cell bodies. The spinal cord at segments 9–10th was exposed for visualizing spinal neurons[40]. The preparation was transferred to a Sylgard-coated recording chamber and positioned ventral side up for easy access to the neurons in MiV1, MiV2 and CaD nuclei in the hindbrain. The dissected spinal segments were placed in a lateral-up position, and two fine tungsten pins were inserted through the notochord at the 6th and 13th spinal segments into the Sylgard layer of the chamber facilitating close proximity of recording electrodes to motoneurons in the spinal cord. To better perform the whole-cell patch-clamp recording, the preparation was further fixed in Vaseline leaving the exposed hindbrain and spinal cord uncovered.

## Behavioral recording in in vivo head-fixed preparation

After a 30-min recovery, asymmetric optic flow to the left or right were presented in front of the in vivo head-fixed preparation at a speed of 0.9 rad/s. Each trial lasted for 1 min and a 10-min interval was applied between trials to prevent habituation. The tail movements were recorded using a high-speed camera (MV-SUA133GC, Mindvision) at 250 fps, outfitted with a lens (XF-MH02X149, Canrill).

## Behavioral analysis

The recorded videos were processed using ZebraZoom[69] to automatically identify and analyze head and tail movements induced by optic flow. The locomotor parameters such as head direction change or tail bending were further characterized using MATLAB scripts. Each bout was identified by locating the peak of the tail angle and its corresponding time (referred to as f1) using the find-peak function. The ten frames preceding f1 (including f1) and thirty frames after f1 were included to define a bout event. For each bout, we obtained the traces of head and tail angle changes over time. Classification was based on the parameters of the tail angle curve, which initially clustered them into rapid steering turn and exploratory routine swimming turn by using an unsupervised Time Series k-means (TSkmeans) algorithm[70], with Dynamic Time Warping (DTW) employed as the distance metric. This preliminary categorization was refined by annotating the bouts derived from the original video recordings. Subsequently, supervised learning techniques were employed using the Classification Learner Application in MATLAB. We utilized the cubic KNN (k-nearest neighbors) learner as it consistently produced the best classification accuracy. We leveraged a cubic KNN learner, incorporating the mean bout angle (defined as the integral of the tail oscillations of the locomotor bout) and tail angle (maximum tail amplitude of the first cycles) as key features for each locomotor bout. This integration of unsupervised classification labels with video annotation benefited the training of the cubic KNN learner, resulting in a classifier. All locomotor bouts in this study were analyzed and then classified using this classifier. On each axis, the probability density function was displayed for the two types of turning movements.

## Electrophysiology

Whole-cell patch-clamp recordings were performed on either GFP-positive motoneurons (MNs) or retrogradely labeled brainstem projecting neurons. The intracellular solution contained (in mM): 120 K-gluconate, 5 KCl, 10 HEPES, 4 Mg$_2$ATP, 0.3 Na$_4$GTP, 10 Na-phosphocreatine (pH 7.4, adjusted with KOH and osmolality 275 mOsm). These neurons were visualized using a fluorescence microscope (Scientifica) equipped with IR-differential interference contrast optics and a CCD camera with frame grabber (QIMAGING). Electrodes for whole-cell recording were advanced into the brain or spinal cord using a motorized micromanipulator (Scientifica) while applying constant positive pressure. Intracellular signals were amplified with a MultiClamp 700B intracellular amplifier (Molecular Devices) and low-pass filtered at 10 kHz. No bias current was injected into the recorded neurons when performing whole-cell recordings. A square current pulse was injected into the recorded neurons to determine their firing threshold. Single short duration current pulses (1 ms) were used to stimulate presynaptic neurons in brainstem nuclei and excitatory postsynaptic potentials (EPSPs) or inhibitory postsynaptic potentials (IPSPs) were recorded in spinal motoneurons. Electrical coupling was tested by injecting positive and negative current pulses among neurons in brainstem nuclei. The excitatory nature of the chemical synaptic transmission was determined by using NBQX (20 μM, Sigma) and APV (50 μM, Sigma) and the inhibitory nature of the chemical synaptic transmission was determined by using strychnine (1 μM, Sigma). The firing threshold of each neuron was determined from the measured membrane potential at which the $dV/dt$ exceeded 10 V/s. A suprathreshold current with a 50 ms duration was

injected into the neurons in the brainstem nuclei while recording the discharging spinal motoneurons. This allowed us to determine whether the neurons in brainstem nuclei increased or decreased the firing of spinal motoneurons. Video recording of tail movement at 250 fps was triggered simultaneously with the electrophysiology recordings by the NI-5734 digitizer (National Instruments) AUX I/O port.

## Single-cell sequencing

Neurons in the CaD nucleus were visualized in the Tg(*Glyt2:GFP*) fish line by retrograde labeling using Dextran Tetramethylrhodamine (3000 MW), collected with a glass pipette, and then transferred to a lysis buffer solution[30]. Total RNA was extracted using the SMART-Seq v4 Ultra Low Input RNA Kit. RNA integrity was assessed using the RNA Nano 6000 Assay Kit of the Bioanalyzer 2100 system (Agilent Technologies, CA, USA). RNA-Seq libraries for each sample were prepared using the NEBNext Ultra RNA Library Prep Kit for Illumina, following the manufacturer's recommendations. The library quality was assessed on the Agilent Bioanalyzer 2100 system. Libraries were sequenced on an Illumina NovaSeq 6000 platform, generating 150 bp paired-end reads for each sample. Raw data of fastq format were firstly processed using in-house Perl scripts. HTSeq v0.6.0 was used to count the read numbers mapped to each gene. Gene expression level was estimated by FPKM (Fragments Per Kilobase of transcript per Million fragments mapped) based on the length of the gene and reads count mapped to this gene.

## Multiplane two-photon calcium imaging

For two-photon calcium imaging, the ventral-up or dorsal-up in vivo preparations were used to detect the calcium response of individual neurons in the brainstem or pretectum. The in vivo preparation was placed under a two-photon microscope (Scientifica) equipped with IR-differential interference contrast optics and a CCD camera with frame grabber (QIMAGING). The calcium responses was measured in brainstem projecting neurons identified by retrograde labeling or neurons in pretectum in Tg(*elavl3:H2B-GCaMP6s x mitfaw2-w2:roya9-a9*). The Tg(*chx10:GCaMP6s*) fish line was also used to measure the activity of pretectal V2a neurons during asymmetric optic flow. We used resonant scanning with a laser at 900 nm for calcium dynamics and 850 nm for retrograde labeling. The imaging area was scanned at four planes with a frame rate of 7–8 volumes per second, and a 30 frames per second (512 × 512 pixels) image was acquired. Images were acquired at 512 × 512 pixels for calcium imaging of the hindbrain and spinal cord. Imaging data were acquired using LabView (National Instruments) and analyzed with custom-written MATLAB scripts. Videos were recorded with a high-speed camera at 250 fps and synchronized with calcium imaging using the NI-5734 digitizer.

## Laser ablation

For the laser ablation experiments, fish were initially anesthetized with 0.03% MS-222 and embedded dorsally up with 3% low-melting agarose (Sigma-Aldrich) in a recording chamber. They were perfused with system water containing 0.03% MS-222[30,40]. The chamber was placed on the stage of a two-photon microscope (Scientifica). A two-photon laser (Coherent laser Ultra II) operating at an 80 MHz repetition rate and a pulse duration of less than 140 fs was applied. The microscope was equipped with IR-differential interference contrast optics and a CCD camera with a frame grabber (QIMAGING) to specifically target and ablate unilateral V2a and V0d neurons in brainstem nuclei or pretectal V2a neurons. The ablation experiments were performed in four ways: (1) unilateral ablation of V0d neurons in CaD only; (2) unilateral ablation of steering V2a neurons in MiV1 and MiV2; (3) unilateral ablation of both steering V2a and V0d neurons in MiV1 and MiV2; (4) unilateral ablation of pretectal V2a neurons. High-power single laser pulses (duration: 10–20 ms) were focused with a 0.8 NA ×40 objective (Olympus) on the above-mentioned neurons. The laser power was set

to approximately 100–200 mW, as measured with a power meter under the objective. The ablation was indicated by brief high-intensity flashes and verified through the permanent loss of fluorescence confirmed using IR-differential interference contrast and a CCD camera. The success of the ablation procedure was rechecked the next day to confirm the loss of fluorescence. After ablation, the animals were released and allowed to recover for 2–4 h in fish water at room temperature. Behavioral experiments were conducted before and after ablation, with both control and ablated groups undergoing two trials each. Behavioral results were analyzed pre- and post-ablation in each fish.

## Immunohistochemistry

To identify the V2a and V0d neurons in MiV1, MiV2 and CaD nuclei, Mauthner cells and their axons were used as spatial references in the Tg(*Tol-056:GFP*) fish line. Neurobiotin (0.25%, Vector Laboratories) was added to the intracellular solution to reveal the morphology of the V2a and V0d neurons in MiV1, MiV2, and CaD nuclei. A higher concentration of neurobiotin (2%, SP-1120, Vector Laboratories) was injected into the whole-cell recorded V2a and V0d neurons in MiV1, MiV2, and CaD nuclei, which allowed us to label the pretectal V2a neurons via dye coupling. A 500 ms duration threshold current stimulation pulse was applied every 30 sec to the recorded neurons for 30–35 min, facilitating the transport of neurobiotin. The preparations were then fixed with 4% paraformaldehyde (PFA) in phosphate-buffered saline (PBS, 0.01 M; pH 7.4). After 24-hour fixation, the brains were dissected and processed using streptavidin conjugated to Alexa Fluor 647 (1:400, S32357, Invitrogen) or Alexa Fluor 555 (1:500, S32355, Invitrogen). The brains underwent five washes in PBS, each lasting 5 min. Non-specific protein binding sites were blocked using a mixture of 3% BSA, 2% normal donkey serum, and 1% Triton X-100 in PBS for 30 min at room temperature. The brains were then incubated with streptavidin in 1% Triton X-100 in PBS at 4 °C for 48 h, followed by rinsing in PBS and mounting on coverslips with anti-fading fluorescent mounting medium (Invitrogen). A laser scanning confocal microscope (FV3000, Olympus) was utilized to examine the tissue and acquire fluorescence images. The morphology of the neurons was manually reconstructed in CorelDraw (Corel, https://www.coreldraw.com/).

## Calcium imaging analysis

For calcium imaging, we used the nacre Tg(*elavl3:H2B-GCaMP6s*) fish or Tg(*Chx10:Gal4;UAS: GCaMP6s*). To analyze the calcium responses of individual neurons, image cross-correlation techniques were initially applied for motion correction. Individual regions of interest (ROIs), each representing a neuron in the pretectum, dorsal thalamus, optic tectum, or brainstem, were manually delineated on the averaged image derived from the entire series. For each ROI, the raw fluorescence intensity ($F$) and background fluorescence intensity ($Fb$) were extracted. $F' = F - Fb$ was calculated, and then $\Delta F/F(0) = (F'(t) - F0)/F0$, where $F'(t)$ represents the fluorescence at time $t$, and $F0$ is the baseline fluorescence of the cell (average fluorescence over a 5-s period of $F'$ inactivity). Gaussian filtering was employed to reduce the noise in the $\Delta F/F(0)$ signal, followed by further smoothing using the Savitzky-Golay filter (sgolayfilt[71], yielding each ROI's peak fluorescence ($Fmax$) and calcium signal amplitude ($Fa = Fmax - F0$). The calcium response curves were synchronized with the high-speed camera recordings. We focused on the pretectal neurons close to the nMLF region, particularly planes at depths of 64-73 μm according to the Z Brain Atlas (https://zebrafishexplorer.zib.de/home/). ROIs in the pretectum with a $Fa$ exceeding 0.3 were defined as responsive neurons ($n1$), based on previously reported evidence suggesting that such a $Fa$ level in GCaMP6s-expressing neurons is indicative of action potential activity[72]. This threshold allowed the effective selection and enumeration of recruited neurons within the pretectum. The total number of neurons in the pretectum was denoted as $n$, with the response rate

calculated as *n1/n*. The normalized *Fa* values of these responsive neurons were subject to statistical analysis.

### Electrical stimulation

We used the dorsal-up in vivo preparation to perform electrical stimulation of the ventral pretectum. The brain was carefully dissected to allow the stimulation electrode to reach the ventral pretectum. Stimulation was delivered using a stimulus isolator (IsoFlex, A.M.P.I., Israel), which together with a high-speed camera was simultaneously triggered by TTL signals from a Digidata series 1550B digitizer (Axon Instruments, Molecular Devices). This arrangement ensured synchronization of electrophysiology and video recording of tail movement. Short-duration pulses (pulse width: 0.2 ms) were utilized to stimulate the ventral pretectum region using a micropipette with a tip diameter of 2–2.5 µm while performing whole-cell recordings of the V2a and V0d neurons in MiV1, MiV2 and CaD nuclei. Four stimulation intensities (1×, 2×, 5×, 10×) were applied to each preparation. The intensity (1×) represented the threshold stimulation to elicit turning movements. A high $Ca^{2+}/Mg^{2+}$ solution (in mM, 109 NaCl, 2.9 KCl, 10 $CaCl_2$, 10 $MgCl_2$, 10 HEPES, and 10 glucose, pH 7.8 adjusted with NaOH, osmolality 290 mOsm) was employed to identify monosynaptic connections.

### Optogenetic stimulation

We used the dorsal-up in vivo preparation to perform optogenetic stimulation of the ventral pretectum in the Tg(*Chx10:Gal4;UAS:ChR2-mCherry*). Whole-cell recordings were first performed in V2a and V0d neurons in MiV1, MiV2 and CaD nuclei, then the objective (×60, 1.0 NA, Olympus) was shifted to the ventral pretectum to deliver the blue light. Optical stimulation was administered using a Polygon 1000 Digital Micromirror Device (Mightex). Simultaneous performance of optogenetic stimulation, electrophysiology and video recording of tail movement were synchronized using TTL signals from a Digidata 1550B digitizer (Axon Instruments, Molecular Devices). A 50 ms light pulse (470 nm) was used for optical stimulation with four intensities (1×, 2×, 5×, 10×). The intensity (1×) was defined as the threshold for eliciting turning movements (intensity: 4.6–5.2 µW).

### Quantification and statistical analysis

Electrophysiological data were analyzed using Spike2 (Cambridge Electronic Design), Clampfit (Molecular Devices), and MATLAB software (Mathworks). The EPSP amplitudes were calculated as the difference between baseline and peak in mV. The recorded EPSPs or IPSPs represent the average of 30–60 consecutive sweeps. The timing of the spikes in the recorded neurons was shown as raster plots in Fig. 5 and were generated in MATLAB. The calcium response curves were analyzed by a MATLAB script. The heat map and contour plot were also generated in MATLAB. Photoshop (Adobe), Fiji[73] (NIH) were used to process images. Significant differences between the means in experimental groups and conditions were analyzed by the Student's *t*-test using Prism 9.0 (GraphPad Software, http://www.graphpad.com/). Differences were considered significant if $P < 0.05$. All data presented are given as mean ± SEM. The number (*n*) given with the mean refers to independent experiments in different animals. All figures were created by the authors using CorelDraw (Corel Corporation).

### Reporting summary

Further information on research design is available in the Nature Portfolio Reporting Summary linked to this article.

## Data availability

All data are available in the main text and Supplementary Information. Source data can be accessed at https://doi.org/10.6084/m9.figshare. 27937887. Single-cell RNA sequencing raw and processed data files are deposited in the NCBI Gene Expression Omnibus (GSE290157) and can be accessed at https://www.ncbi.nlm.nih.gov/geo/query/acc.cgi?acc= GSE290157.

## Code availability

All code used in this study are available at https://doi.org/10.6084/m9. figshare.27937887.

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

## Acknowledgements

We thank Dr. S Grillner for comments on the study; Dr. J-L Du and China Zebrafish Resource Center (CZRC) for sharing the fish lines; J Mao and C Wang for fish line generation and care. This work was supported by STI2030-Major Projects (2021ZD0204500 and 2021ZD0204501), the National Natural Science Foundation of China (32320103004, 32300831 and 32271051), Shanghai Pilot Program for Basic Research and Shanghai Rising-Star Program (23YF1449900). A. El Manira is supported by a European Research Council Advanced Grant (101142013), the Swedish Research Council, Sweden (2017-02905); Knut and Alice Wallenberg Foundation, Sweden (KAW 2018.0010 and KAW 2022.0130).

## Author contributions

L. Xu performed most of the experiments and analysis with help from B. Zhu, X. Tao, Z. Zhu and T. Zhang. J. Song and A. El Manira designed the experiments, interpreted the results and wrote the manuscript.

## Funding

## Competing interests

The authors declare no competing interests.
