## [Transparent Peer Review file · Nature Communications]

Separate brainstem circuits for fast steering and slow exploratory turns

Corresponding Author: Professor Abdel El Manira

Version 0:

Reviewer comments:

Reviewer #1

(Remarks to the Author)

This study identifies separate populations of brainstem neurons responsible for rapid steering and for slow exploratory turns using a combination of electrophysiology, optogenetics, retrograde tracing, calcium imaging, laser ablation and behavioural analysis. The results represent a major advance in our understanding of the neural circuits that control movement, in this case, identifying precisely the brainstem nuclei and the neurons that project commands to spinal circuits to control specific facets of locomotion.

The experiments are well designed, using a wide array of tools and techniques that harness the larval zebrafish model, and the results provide very strong support for their conclusions.

I only have minor comments

1. Figure 1J-K. Can you show supplementary data showing that the ventral side-up positioning of the fish does not affect any of the conclusions made?
2. Fig. 2e and f. Were there post-hoc adjustments for the paired t-tests?
3. Fig 3. Did one cell precede another? It looks from the patch-clamp recordings that V0d-CaD may precede V2a-MiV1
4. In the Results, the sentence "However, pretectal V2a neurons were bilaterally and symmetrically recruited during exploratory rou_ne turns (Fig. 6f, g)." It looks like Fig. 6f, g is not the correct figure citation

Reviewer #2

(Remarks to the Author)

The paper by Lu et al. identifies and characterizes the role of distinct brainstem circuits for fast and slow steering during zebrafish swimming. The authors document clearly that they can generate and distinguish fast from slow turnings during an optokinetic task. Using calcium imaging in head-fixed preparations, they show that distributed spinal projecting brainstem neurons are unilaterally activated during left versus right fast and slow turnings. They show that the firing pattern of brainstem V2a neurons is correlated to the amplitude and not the frequency of the tail bending, and that V2a neurons are electrically coupled to V0d neurons located in the caudal brainstem. Ablation of either or both V2a or V0d neurons of three brainstem nuclei impairs turning. Regarding functional locomotor control, they also show elegantly that excitatory V2a neurons control ipsilateral motoneurons, whereas inhibitory V0d neurons inhibit contralateral motoneurons. Finally, they show that the activity of pretectum V2a neurons is modulated as a function of the left versus right optokinetic test. Finally, photoactivation of pretectum V2a neurons drives activity in downstream brainstem V2a and V0d neurons and induces ipsilateral turning. Conversely, ablation of these pretectum V2a neurons impairs ipsilateral turns. Overall, it is a very good study, but there are minor weaknesses that should be easy to address.

Strengths:

The figures are clear and well described.

The study combines elegant techniques including calcium imaging, neural recording, electrical and optogenetic stimulation, and laser ablation.

Weaknesses:

My biggest concern is the insistence of the authors to refer to slow turnings as “exploration” throughout the draft, while there is no evidence or data supporting that. What is an explorative turn in comparison to a steering turn? Is it just an angular amplitude of less than 20 degrees? Are there really two distinct states controlled by two different circuits? From examples provided (Fig. 2 and extended data), it looks like there is a continuum rather than distinct states. Although less activated, all neurons are activated during slow and fast turning. The “explorative turn” actually looks like passive drifting. Circuits controlling explorative locomotion appears embedded in MiV1 and MiV2 among neurons controlling a steering turn. The authors need to better demonstrate the existence of two distinct circuits and states.

It is surprising that Neurobiotin can pass through gap junctions of remote neurons. Usually, gap junctions are between adjacent neurons of the same type, allowing synchronization of a specific population. The electrophysiological demonstration in Fig. 3c and 3f is convincing and therefore puzzling. The intensity of Neurobiotin in remote neurons is really high and could suggest a spillover in the extracellular space. Neurobiotin could have been recaptured by axon terminals of V2a or V0d neurons. How far apart are these neurons? Do the authors have studies in mind showing connectivity between remote neurons? What would be the types of junctions (axo-somatic, axo-axonic, dendro-dendritic, or any other combination)? What would be the advantage of a long-distance electrical coupling over chemical synapses organized in a serial fashion? Moreover, the authors discuss gap junctions involved in coupling brainstem neurons to generate a sharp steering angle. Gap junctions should be blocked to support this claim.

Regarding other mouse studies, it might worth discussing the study by Usseglio et al., 2020. In this study, they showed that if unilateral optogenetic activation of cervical-projecting brainstem V2a neurons induces head turning, activation of lumbar-projecting brainstem ones appears to systematically stop locomotion in mammals. This is in striking contrast with the current study, which argues that long-distance-projecting V2a neurons generate turnings and not stops in zebrafish.

Minor comments

Abstract

Introduction

The introduction would gain clarity if the penultimate paragraph describing the results focused on the neural circuits controlling fast then slow turning, rather than describing them in the chronological order of presentation in the results section. If V2a neurons excite motoneurons, V0d neurons inhibit contralateral motoneurons; therefore both cell types cannot ensure a recruitment of fast primary motoneurons on both sides.

Results

There is no exploration in Fig. 1: it is a fast or slow turning triggered by an optokinetic reflex. Steering versus exploratory turns should be corrected throughout the draft to fast versus slow turns.

Page 7, 2nd paragraph: Correct 40 degrees.

How did the authors ensure that both neurons were electrically coupled?

Page 8 “heading direction.” Heading and direction are synonymous. However, the title should mention that this is about “change of heading” or “changing of direction.” This occurs throughout the text.

Fig. 2b and e: It is not necessarily clear that these are distinct neurons. It is very difficult to appreciate the narrative with the overlapping of color-coded traces. A heatmap identifying individual neurons as rows would increase readability and appreciation of the data.

Fig 2c, f: For each nucleus, there are two groups to perform paired statistics. What is the condition tested? It should be clearly stated in the legend and results. Also, would it be possible to number the neurons within a nucleus? The current bar graph is cumbersome.

Figure 3: I suggest rephrasing the title. This not a recruitment but rather the firing pattern of steering neurons, and this firing is not dependent on but rather correlated to tail-bending amplitude.

Extended Fig 3: Panel d is not referenced in the legend. There is a typo for “angles” on the x-axis. This typo occurs in several figures.

Figure 5: How were ultra-fast motoneurons identified? This should be briefly mentioned in the results. Concerning monosynaptic connectivity, please report the latency.

Fig 6f: I don't understand the link between an increase of firing with light intensity and the text about bilateral and symmetrical recruitment. This needs clarification.

Page 10, end of 2nd paragraph: “...brainstem steering circuits via synaptic connections,” I guess that the authors are referring to electrical synapse?

Discussion

Page 11, 2nd paragraph: "shallow turns associated with exploratory swimming." This is a slow turn; there is no evidence that it is associated with exploration. This should be corrected throughout the text.

Page 12, 2nd paragraph: "the recruitment of these steering... neurons is influenced... loco frequency." No, it is correlated not influenced.

In the discussion about the two circuits, there is one that adjusts its activity to control amplitude of movement and the other is a two-module circuit. This is not clear.

Reviewer #3

(Remarks to the Author)

This is a thorough study using behavioral measurements, electrophysiology, genetics, tracing, and imaging to determine visual flow through sets of V2a circuits at the pretectum and the brainstem, which establish connections to the motor neurons in the spinal cord that control the directionality of swimming.

I have a few suggestions that hopefully will improve the manuscript.

Conceptual/major points

- Fig. 1i: It is unclear how/why the asymmetric optic flow induces fast steering turns. What is the physiological relevance?

Adding a sentence to explain what the asymmetric optic flow represents would be helpful. I would think that something like looming stimuli might be much more relevant and closely reflect what fish might experience in the wild.

- Fig 2 c and f. $\Delta F/F$ are variable among fish. Since the electrophysiology data in the latter part of the paper is so clear-cut (Fig 3), I am wondering if there is a correlation with modulation of activity and behavior, i.e., fish that exhibit small $\Delta F/F$ show, in general, smaller tail or head angles during steering turns than fish that exhibit large $\Delta F/F$. Please show this.

- Regarding the link between direction selectivity and steering turns (Fig 6, Ext Fig 5-7): Perhaps because I did not understand the physiological relevance of asymmetric optical flow, the logic of experimental design here is difficult to follow. For fast escape, which would require a steering turn, I would think that fish will care about not only which direction visual input is flowing but also where the visual field the input appears, size, speed of movement, etc, that might be processed by thalamus and optic tectum. In other words, it's fine that authors show that the pretectum represents direction selectivity, but would the authors get the same behavioral outcome, fast steering movement, by stimulating the thalamus and/or optic tectum, driven by other types of visual information? Description of what the authors think the visual flow represents and that Fig 6 is focused on a sub-feature of visual information that drives the circuit recruitment and behavior is appreciated.

Minor points:

- Fig 2. B and e: Labels here are written as "braistem" instead of brainstem.

- Ext Data 6d: labels are spelled "angel" :)

Version 1:

Reviewer comments:

Reviewer #1

(Remarks to the Author)

The authors have addressed my comments very well. I have no further issue with the manuscript.

Reviewer #2

(Remarks to the Author)

The authors have addressed my main concerns.

The existence of distinct pathways for fast versus slow turns is convincing. Since all neurons appear to be identified across preparations, they could be labeled (with a number or another identifier) for clearer reference.

Since there are no behavioral analyses in the current study, using the term "exploration" to describe slow turns is an overinterpretation as it is based solely on kinematic measurements. The authors cite several references suggesting that slow turns are more "associated" with exploration, but it cannot be ruled out that sharp turns may also occur during exploration. External stimuli, beyond just optical flow, could also contribute to generating such sharp exploratory turns especially in an open environment. The association of slow turns with exploration should be clarified or disclaimed as a paradigm in the introduction. An ethogram in Fig. 1b could also help illustrate the concept of two distinct turning behaviors.

Thank you: the heatmaps are much clearer than the color-coded traces in the current Fig. 2. Since the authors wish to keep the original Fig. 2, the heatmaps should be included as supplementary figures.

Page 4: For consistency, the past tense should be used when describing the updated results.

Reviewer #3

(Remarks to the Author)

I am happy with the revised version.

Response to Reviewers

Reviewer #1:

This study identifies separate populations of brainstem neurons responsible for rapid steering and for slow exploratory turns using a combination of electrophysiology, optogenetics, retrograde tracing, calcium imaging, laser ablation and behavioural analysis. The results represent a major advance in our understanding of the neural circuits that control movement, in this case, identifying precisely the brainstem nuclei and the neurons that project commands to spinal circuits to control specific facets of locomotion.

The experiments are well designed, using a wide array of tools and techniques that harness the larval zebrafish model, and the results provide very strong support for their conclusions.

Response: We thank the reviewer for their positive comments.

I only have minor comments

1. Figure 1J-K. Can you show supplementary data showing that the ventral side-up positioning of the fish does not affect any of the conclusions made?

Response: There was no significant difference in the features of turns between the ventral side-up and dorsal side-up preparations, as shown in the new Extended Data Fig. 1. The patterns and parameters of turning behavior in the ventral side-up preparation were similar to those in the dorsal side-up preparation (Extended Data Fig. 1a). These data are now described in the text on page 5, lines 115-117.

2. Fig. 2e and f. Were there post-hoc adjustments for the paired t-tests?

Response: No post-hoc adjustments were applied for the paired t-tests.

3. Fig 3. Did one cell precede another? It looks from the patch-clamp recordings that V0d-CaD may precede V2a-MiV1

Response: We have analyzed the onset time of the first spike in relation to the timing of the large tail angle in the V0d- CaD and V2a-MiV neurons. There was no significant difference in the timing of the spike between the two classes of neurons. This analysis is shown in the graph below. We have also indicated this in the text on page 7, lines 179-181.

4. In the Results, the sentence "However, pretectal V2a neurons were bilaterally and symmetrically recruited during exploratory routine turns (Fig. 6f, g)." It looks like Fig. 6f, g is not the correct figure citation

Response: This has been corrected – thank you!

Reviewer #2:

The paper by Lu et al. identifies and characterizes the role of distinct brainstem circuits for fast and slow steering during zebrafish swimming. The authors document clearly that they can generate and distinguish fast from slow turnings during an optokinetic task. Using calcium imaging in head-fixed preparations, they show that distributed spinal projecting brainstem neurons are unilaterally activated during left versus right fast and slow turnings. They show that the firing pattern of brainstem V2a neurons is correlated to the amplitude and not the frequency of the tail bending, and that V2a neurons are electrically coupled to V0d neurons located in the caudal brainstem. Ablation of either or both V2a or V0d neurons of three brainstem nuclei impairs turning. Regarding functional locomotor control, they also show elegantly that excitatory V2a neurons control ipsilateral motoneurons, whereas inhibitory V0d neurons inhibit contralateral motoneurons. Finally, they show that the activity of pretectum V2a neurons is modulated as a function of the left versus right optokinetic test. Finally, photoactivation of pretectum V2a neurons drives activity in downstream brainstem V2a and V0d neurons and induces ipsilateral turning. Conversely, ablation of these pretectum V2a neurons impairs ipsilateral turns. Overall, it is a very good study, but there are minor weaknesses that should be easy to address.

Strengths:

The figures are clear and well described.

The study combines elegant techniques including calcium imaging, neural recording, electrical and optogenetic stimulation, and laser ablation.

Response: We thank the reviewer for the positive comments.

Weaknesses:

My biggest concern is the insistence of the authors to refer to slow turnings as “exploration” throughout the draft, while there is no evidence or data supporting that. What is an explorative turn in comparison to a steering turn? Is it just an angular amplitude of less than 20 degrees? Are there really two distinct states controlled by two different circuits? From examples provided (Fig. 2 and extended data), it looks like there is a continuum rather than distinct states. Although less activated, all neurons are activated during slow and fast turning. The “explorative turn” actually looks like passive drifting. Circuits controlling explorative locomotion appears embedded in MiV1 and MiV2 among neurons controlling a steering turn. The authors need to better demonstrate the existence of two distinct circuits and states.

Response: We appreciate this question and provide the following evidence supporting our conclusion that fast steering turns and slow exploratory turns are mediated by distinct circuits in the brainstem, which in turn connect to different motoneuron types in the spinal cord.

Figure 1 presents an analysis of turn angles, clearly distinguishing between two types of turns based on changes in tail angle and the corresponding changes in heading direction. These turn types can be differentiated based on the distribution of tail angles.

Calcium imaging and electrophysiological recordings reveal that specific neurons are selectively active during either fast, large steering turns or slow, shallow exploratory turns. Even within the same nucleus, distinct populations of V2a neurons encode steering and exploratory turns. The data presented in Figure 2 show that fast and slow turns are encoded by separate neuronal populations located in different brainstem nuclei.

Selective ablation of neurons responsible for encoding fast steering turns resulted in the complete loss of these turns, with no impact on slow exploratory turns (Fig. 4i-l). Conversely, the selective ablation of brainstem neurons that encode slow turns significantly decreased the occurrence of slow exploratory turns while leaving steering turns unaffected (new Extended Data Fig. 5k-o).

Finally, we show that neurons responsible for encoding fast steering turns are connected to fast primary motoneurons in the spinal cord, providing strong excitation to ipsilateral motoneurons while simultaneously exerting strong inhibition on contralateral motoneurons. In contrast, neurons that encode exploratory turns are connected to slow secondary motoneurons in the spinal cord (see new Extended Data Fig. 5a-j).

Together, these data demonstrate that brainstem neurons and their target spinal motoneurons form distinct circuits that selectively encode fast steering turns and slow exploratory turns.

We have revised the results (page 4, lines 89-95; page 5, lines 102-104) and discussion to describe these points to prevent potential confusion (page 12, lines 330-333; page 12, lines 342-347).

It is surprising that Neurobiotin can pass through gap junctions of remote neurons. Usually, gap junctions are between adjacent neurons of the same type, allowing synchronization of a specific population. The electrophysiological demonstration in Fig. 3c and 3f is convincing and therefore puzzling. The intensity of Neurobiotin in remote neurons is really high and could suggest a spillover in the extracellular space. Neurobiotin could have been recaptured by axon terminals of V2a or V0d neurons. How far apart are these neurons? Do the authors have studies in mind showing connectivity between remote neurons? What would be the types of junctions (axo-somatic, axo-axonic, dendro-dendritic, or any other combination)? What would be the advantage of a long-distance electrical coupling over chemical synapses organized in a serial fashion? Moreover, the authors discuss gap junctions involved in coupling brainstem neurons to generate a sharp steering angle. Gap junctions should be blocked to support this claim.

Response: We have extensive experience with neurobiotin dye-coupling in the spinal cord and brainstem of zebrafish. Additionally, the laboratory of Shin-ichi Higashijima has shown the existence of neurobiotin dye-coupling from a single Mauthner cell in the brainstem to specific

inhibitory commissural interneurons (called CoLo) in the spinal cord, over a distance exceeding 600 μm (Figure 4 in Satou et al., 2009). Our own studies in the adult zebrafish spinal cord corroborate these findings (Song et al, 2016; PMID: 26760208). We have used a high concentration of neurobiotin to load single motoneurons, resulting in dye-coupling with several V2a interneurons located 2-3 spinal segments away from the recorded motoneurons (a distance of over 300 μm ; Figure 1, Song et al., 2016). Therefore, dye-coupling through axon-dendritic gap junctions occurs in both adult and larval zebrafish, allowing us to exclude any non-specific labeling or potential spillover of neurobiotin into the extracellular space.

Furthermore, we have now performed additional experiments using paired whole-cell recordings from V2a/V0D neurons and ventral pretectal V2a interneurons. The results show the presence of bidirectional electrical coupling between these neurons (Extended Data Fig. 8d-e). In accordance with the reviewer's suggestion, we have used a gap junction blocker, Carbenoxolone, which completely eliminated the electrical component (Extended Data Fig. 8e). We have revised the text to incorporate these additional experimental data (pages 10-11, lines 288-295).

Regarding other mouse studies, it might worth discussing the study by Usseglio et al., 2020. In this study, they showed that if unilateral optogenetic activation of cervical-projecting brainstem V2a neurons induces head turning, activation of lumbar-projecting brainstem ones appears to systematically stop locomotion in mammals. This is in striking contrast with the current study, which argues that long-distance-projecting V2a neurons generate turnings and not stops in zebrafish.

Response: Thank you for the suggestion. This study is relevant, and we have discussed it in the context of our findings and other studies conducted on mice. It is clear that there is a conserved role of brainstem V2a neurons in controlling turning and changes in heading direction. Our results do not rule out the involvement of brainstem V2a neurons in halting locomotion, as was shown in mice. We have revised the discussion accordingly and have followed the reviewer's suggestions (page 15, lines 425-430).

Minor comments

Abstract

Introduction

The introduction would gain clarity if the penultimate paragraph describing the results focused on the neural circuits controlling fast then slow turning, rather than describing them in the chronological order of presentation in the results section.

If V2a neurons excite motoneurons, V0d neurons inhibit contralateral motoneurons; therefore both cell types cannot ensure a recruitment of fast primary motoneurons on both sides.

Response: Thank you for pointing this out. This has been corrected (Page 4, Lines 70-71).

Results

There is no exploration in Fig. 1: it is a fast or slow turning triggered by an optokinetic reflex. Steering versus exploratory turns should be corrected throughout the draft to fast versus slow turns.

Response: A large and fast change in tail angle is associated with rapid change in heading direction observed during steering. In contrast, slow and shallow changes in tail angle are typically seen during exploratory behavior, as previously reported (Dunn et al., 2016, PMID: 27003593; Marques et al., 2018, PMID: 29307558). The use of optic flow has been used as a means to induce these two patterns of turns and to examine their underlying neural circuits. We prefer to focus on the behavioral correlates of these turns rather than their kinematic features; therefore, we would like to maintain the distinction between steering and exploratory turns. We hope the reviewer agrees.

We have revised the introduction and results sections to clarify this (page 4, lines 89-95; page 5, lines 102-104).

Page 7, 2nd paragraph: Correct 40 degrees.

Response: Corrected, thank you!

How did the authors ensure that both neurons were electrically coupled?

Response: We performed paired whole-cell patch-clamp recordings between these neurons to confirm the existence of bidirectional electrical coupling between MiV-V2a and CaD-V0d neurons (Fig. 3c and f).

Page 8 “heading direction.” Heading and direction are synonymous. However, the title should mention that this is about “change of heading” or “changing of direction.” This occurs throughout the text.

Response: This has been corrected throughout the manuscript.

Fig. 2b and e: It is not necessarily clear that these are distinct neurons. It is very difficult to appreciate the narrative with the overlapping of color-coded traces. A heatmap identifying individual neurons as rows would increase readability and appreciation of the data.

Response: We believe that the data presented offer a more accurate description of the activity of the different neurons and we would like to retain Figure 2 in its current form, subject to the reviewer's approval. We have revised the figure legend to clarify what different color-coded traces. However, we have included the heatmaps for the reviewer's assessment and can provide them as an Extended Data Figure if deemed necessary.

Fig 2c, f: For each nucleus, there are two groups to perform paired statistics. What is the condition tested? It should be clearly stated in the legend and results. Also, would it be possible to number the neurons within a nucleus? The current bar graph is cumbersome.

Response: The statistical comparison of DF/F between the left and right pairs of neurons in the different brainstem nuclei that are active during steering (magenta) versus exploratory (cyan) turns is presented. In the graph, the solid color represents the ipsilateral neurons, while the faint color indicates the contralateral neurons. This distinction is now clearly explained in the figure legend.

Figure 3: I suggest rephrasing the title. This not a recruitment but rather the firing pattern of steering neurons, and this firing is not dependent on but rather correlated to tail-bending amplitude.

Response: We have change “recruitment” to “firing pattern”.

Extended Fig 3: Panel d is not referenced in the legend. There is a typo for “angles” on the x-axis. This typo occurs in several figures.

Response: These have been corrected, thank you!

Figure 5: How were ultra-fast motoneurons identified? This should be briefly mentioned in the results. Concerning monosynaptic connectivity, please report the latency.

Response: The ultra-fast motor neurons refer to the primary motor neurons located in the spinal cord. These neurons possess the largest cell bodies in the spinal cord and can be easily identified under a microscope. There are three to four primary motor neurons in each spinal hemi-segment. In this study, we also used a fish line that specifically marks these primary motoneurons. This information has been incorporated into the methods section (Pages 18, Lines 496-498)..

The average synaptic delay between steering brainstem neurons and pMNs was 0.8 ms, which is typical for monosynaptic connections. This is now stated in the results (Page 9, Lines 228-230).

Fig 6f: I don't understand the link between an increase of firing with light intensity and the text about bilateral and symmetrical recruitment. This needs clarification.

Response: During the steering turns, the ipsilateral pretectal V2a neurons exhibited significantly larger calcium responses compared to their contralateral counterparts, indicating that unilateral activation of pretectal V2a neurons induces ipsilateral steering turns. Neurons in the optic tectum and thalamus were equally activated on both sides, suggesting that the asymmetrical activation of pretectal V2a neurons can lead to ipsilateral steering turns. Indeed, there was an increase in the number of action potentials in pretectal V2a neurons with increasing intensity of unilateral optogenetic stimulation, resulting in a large tail bending amplitude that underlies steering turns.

We now correct the text to clarify the description of Fig. 6f (page 11, lines 304-304).

Page 10, end of 2nd paragraph: “...brainstem steering circuits via synaptic connections,” I guess that the authors are referring to electrical synapse?

Response: We now provide additional experimental data obtained through paired whole-cell patch-clamp recordings, demonstrating that pretectal neurons form mixed electrical and chemical synapses onto steering turn V2a neurons. This data is included in Extended Data Fig. 8d, e and described in the text on pages 10-11, lines 288-295.

Discussion

Page 11, 2nd paragraph: “shallow turns associated with exploratory swimming.” This is a slow turn; there is no evidence that it is associated with exploration. This should be corrected throughout the text.

Response: During slow swimming, zebrafish explore their environment through a sequence of trajectory changes mediated by slow/shallow turns. These exploratory turns can be observed in live animals and may be triggered by asymmetrical optical flow. A similar experimental paradigm was used in previous studies, and therefore, we prefer to relate the induced turns to their behavioral context. The use of exploratory turns has been described in two previous studies in larval zebrafish (Dunn et al., 2016, PMID: 27003593; Marques et al., 2018, PMID: 29307558). We hope the reviewer agrees with our preference for a term that describes behavior rather than merely the kinematics of movement. We have added a few sentences to clarify this point and have cited the two relevant papers in the results (Page 4, Lines 89-95) as well as in the discussion (Page 12, Lines 330-333 and Lines 342-347).

Page 12, 2nd paragraph: “the recruitment of these steering... neurons is influenced...loco frequency.” No, it is correlated not influenced.

Response: This has been revised, thank you!

In the discussion about the two circuits, there is one that adjusts its activity to control amplitude of movement and the other is a two-module circuit. This is not clear.

Response: We rephrase these sentences to clarify this (Page 12, Lines 342-347).

Reviewer #3:

This is a thorough study using behavioral measurements, electrophysiology, genetics, tracing, and imaging to determine visual flow through sets of V2a circuits at the pretectum and the brainstem, which establish connections to the motor neurons in the spinal cord that control the directionality of swimming.

Response: Thank you for the comments.

I have a few suggestions that hopefully will improve the manuscript.

Conceptual/major points

- Fig. 1i: It is unclear how/why the asymmetric optic flow induces fast steering turns. What is the physiological relevance? Adding a sentence to explain what the asymmetric optic flow represents would be helpful. I would think that something like looming stimuli might be much more relevant and closely reflect what fish might experience in the wild.

Response: Thank you for the suggestion. We have incorporated the reviewer's recommendation by adding a description to clarify the representation of asymmetric optic flow (Page 4, Lines 93-95; Page 14, Lines 394-398) in the Results and Discussion sections, respectively. Additionally, we have cited two relevant papers (Bremmer, 2008; Masoner and Hajnal, 2023) accordingly.

The movement of a body through an environment generates a flow of images from the external world across the retina, typically called 'optic flow'. Optic flow provides observers with critical information about their actions in relation to their surroundings, which is essential for effectively controlling the direction of self-motion in any species (Bremmer, 2008; Masoner and Hajnal, 2023). Consequently, asymmetrical optic flow stimuli create an environment that prompts animals to continuously adjust their heading direction during self-motion (Bremmer, 2008; Masoner and Hajnal, 2023). Animals use both rapid steering turns and slow exploratory turns to achieve changes in heading direction in response to asymmetrical optic flow. Thus, animals can flexibly adjust their heading direction by executing either fast, large changes or slow, small changes in response to the asymmetrical optic flow.

In this study, we aim to decipher the brainstem-spinal cord locomotor circuits that control two distinct turning patterns: fast steering turns and slow exploratory turns. The optimal approach to address this question is to induce both turning patterns randomly and independently using a single type of reliable stimulation (asymmetrical optical flow). This allowed us to systematically compare and identify the brainstem neurons specifically for either fast steering turns or slow exploratory turns in the same experimental setup under consistent visual stimuli. We have explored several methods of visual stimulation, including looming, moving dots, and optic flow. Generally, asymmetrical optic flow was the most reliable stimulus.

- Fig 2 c and f. $\Delta F/F$ are variable among fish. Since the electrophysiology data in the latter part of the paper is so clear-cut (Fig 3), I am wondering if there is a correlation with modulation of activity and behavior, i.e., fish that exhibit small $\Delta F/F$ show, in general, smaller tail or head angles during steering turns than fish that exhibit large $\Delta F/F$. Please show this.

Response: Thank you for the suggestion. We have included a new graph (Extended Data Fig. 3k) to show the linear relationship between the amplitude of the calcium response in brainstem steering neurons and the tail angles of the steering turns. Similar to the electrophysiological results (Fig. 3), the steering brainstem neurons showed increased calcium responses, which were accompanied by a large change in the tail angles. We have also included text to describe this (Page 7, Lines 185-188).

- Regarding the link between direction selectivity and steering turns (Fig 6, Ext Fig 5-7): Perhaps because I did not understand the physiological relevance of asymmetric optical flow, the logic of experimental design here is difficult to follow. For fast escape, which would require a steering turn, I would think that fish will care about not only which direction visual input is flowing but also where the visual field the input appears, size, speed of movement, etc, that might be processed by thalamus and optic tectum. In other words, it's fine that authors show that the pretectum represents direction selectivity, but would the authors get the same behavioral outcome, fast steering movement, by stimulating the thalamus and/or optic tectum, driven by other types of visual information? Description of what the authors think the visual flow represents and that Fig 6 is focused on a sub-feature of visual information that drives the circuit recruitment and behavior is appreciated.

Response: We have followed the reviewer's suggestion and now describe the representation of visual flow (Page 4, Lines 94-95; Page 14, Lines 394-398 and 416-419). Additionally, we have included a description stating that the pretectal neurons driving the brainstem neurons responsible for steering turns are a sub-feature of visual information (last sentence in the results; Page 12, Lines 323-325).

Minor points:

- Fig 2. B and e: Labels here are written as "braistem" instead of brainstem.

Response: Corrected, thank you.

- Ext Data 6d: labels are spelled "angel" :)

Response: Corrected, thank you.

A point-by-point response to the reviewer #2

The authors have addressed my main concerns.

Thank you!

The existence of distinct pathways for fast versus slow turns is convincing. Since all neurons appear to be identified across preparations, they could be labeled (with a number or another identifier) for clearer reference.

Neurons encoding rapid steering versus slow turns can be unambiguously identified by their topographic location in different nuclei. Since we have no other way to identify these neurons, we prefer not to assign arbitrary numbers, which could cause confusion.

Since there are no behavioral analyses in the current study, using the term “exploration” to describe slow turns is an overinterpretation as it is based solely on kinematic measurements. The authors cite several references suggesting that slow turns are more “associated” with exploration, but it cannot be ruled out that sharp turns may also occur during exploration. External stimuli, beyond just optical flow, could also contribute to generating such sharp exploratory turns especially in an open environment. The association of slow turns with exploration should be clarified or disclaimed as a paradigm in the introduction. An ethogram in Fig. 1b could also help illustrate the concept of two distinct turning behaviors.

Rapid steering turns are primarily used for hunting, avoidance, or other contexts requiring a quick change in swimming direction. In open environments, exploration is mostly characterized by shallow routine turns. While we cannot exclude rare instances of rapid turns during exploration, they appear to be uncommon.

Thank you: the heatmaps are much clearer than the color-coded traces in the current Fig. 2. Since the authors wish to keep the original Fig. 2, the heatmaps should be included as supplementary figures.

As the reviewer suggested, we have now included this heatmap as Supplementary Fig. 2.

Page 4: For consistency, the past tense should be used when describing the updated results.

Thank you! We have now changed this part from the present tense to the past tense.